# Copula-SVI: Vine-Copula Variational Inference with Stein Refining for Instance-Level Correlation Capturing

**Junxi Xiao** [1]  **Qinliang Su** [1 2]

## Abstract

Mean-field variational inference (VI) is scalable, but its independence assumption can severely limit inference when the posterior is inherently coupled across instances especially for correlated data. Existing structured VI approaches either impose simple dependence patterns or incur substantial cost as dependence becomes richer, leaving efficient higher-order instance-level dependence modeling largely unresolved. We propose **Copula-SVI**, which augments amortized marginals with an explicit vine-copula posterior and refines joint samples with Stein updates toward the true posterior. The vine construction makes dependence learning and sampling practical by decomposing it into bivariate copula factors, enabling edge-minibatched training with variance-aware level-wise sampling and efficient dependence-aware initialization via a sparse vine built from the same sampled edges. Experiments on constrained clustering and time series modeling show consistent improvements over strong structured VI baselines and demonstrate efficient higher-order instance-level dependence modeling.

## 1. Introduction

Variational inference (VI) is widely used to approximate intractable posteriors in latent-variable models (Blei et al., 2017; Kingma & Welling, 2014). In the standard i.i.d. setting, one typically assumes conditional independence across instances, so that $p_{\boldsymbol{\theta}}(\mathbf{X}, \mathbf{Z}) = \prod_{i=1}^{N} p_{\boldsymbol{\theta}}(\mathbf{x}_i \mid \mathbf{z}_i) \, p(\mathbf{z}_i)$. Under this factorization, the exact posterior also factorizes across instance $i$, and the amortized mean-field family $q_{\boldsymbol{\phi}}(\mathbf{Z} \mid \mathbf{X}) = \prod_i q_{\boldsymbol{\phi}}(\mathbf{z}_i \mid \mathbf{x}_i)$ is a natural choice for scalable inference. Many real datasets, however, contain explicit relations among instances, and these relations couple the latent variables together. For example, in time series, structured priors such as $p(\mathbf{z}_{1:N}) = p(\mathbf{z}_1) \prod_{i>2} p(\mathbf{z}_i \mid \mathbf{z}_{i-1})$ induce posterior dependence across time. In graph-structured or relational data, likelihood factors of the form $\prod_{(i,j)\in\mathcal{E}} p_{\boldsymbol{\theta}}(x_{ij} \mid \mathbf{z}_i, \mathbf{z}_j)$ couple latents across nodes through observed edges. In these correlated settings, an instance-factorized posterior approximation is no longer reasonable, as it enforces independence, failing to represent the cross-instance dependencies defining the true posterior (Turner & Sahani, 2011).

Structured VI aims to relax mean-field by introducing dependencies into $q(\mathbf{Z} \mid \mathbf{X})$ (Hoffman & Blei, 2015), and most existing designs primarily target correlations among latent *dimensions* within each $\mathbf{z}_i$. However, the number of latent dimensions is typically small (e.g., dozens to hundreds), while the number of instances in a correlated batch can be much larger (e.g., thousands to millions). Thus, methods modeling latent dimension correlations cannot directly be used to capture correlations among instances, particularly for large batches. Recent instance-level structured VI methods address this by imposing explicit dependence structures across instances. CVAE (Tang et al., 2019) and TreeVI (Xiao & Su, 2024) are scalable but restrict the posterior to a tree, which precludes cycles and limits interactions to first-order effects. HoT-VI (Xiao et al., 2025) increases expressiveness with higher-order factors, but its cost grows rapidly with the dependency order, making long-range or high-order correlations difficult to handle in practice. Moreover, tractability in these approaches relies on restrictive assumptions on marginal/conditional distributions, which could underrepresent non-Gaussian marginals and complex dependence patterns. To overcome the restrictions in variational inference relying on specific forms of distributions, Stein variational gradient descent (SVGD) refines particles toward the posterior via functional gradients (Liu & Wang, 2016), and latent-space SVGD improves stability by performing refinement in the latent space (Pu et al., 2017). However, for correlated data, most latent-space SVGD pipelines still start from an instance-factorized approximation and project back to the same factorized family. This effectively hard-codes independence in the explicit posterior, and any cross-instance

[1] School of Computer Science and Engineering, Sun Yat-sen University, Guangzhou, China [2] Guangdong Key Laboratory of Big Data Analysis and Processing, Guangzhou, China. Correspondence to: Qinliang Su <suqliang@mail.sysu.edu.cn>.

*Proceedings of the 43rd International Conference on Machine Learning*, Seoul, South Korea. PMLR 306, 2026. Copyright 2026 by the author(s).

dependence induced by SVGD is erased as soon as the particles are projected back. The result is a "refine-and-forget" loop that keeps snapping the approximation back to independence, so instance coupling stays weak and cannot accumulate across iterations.

To overcome the problems above, we propose **Copula-SVI**, which breaks this loop by making the approximate posterior able to model complex dependence itself and using Stein refined particles to further improve the dependence modeling rather than overwrite it as in most existing SVGD methods. This is mainly achieved by the introducing of a family of copula-augmented *variational* posteriors, which are able to effectively decouple the marginal distribution modeling from the more difficult dependence modeling. With the copula-augmented posterior, we draw coupled particles, refine them toward the posterior with a few SVGD steps, and fit the same copula-augmented family back to the refined particles. This preserves and strengthens the learned dependence, avoiding the factorized projection in current latent-space SVGD methods that would otherwise erase it. To scale up the dependence learning to large datasets, we instantiate the copula as a regular vine, which factorizes high-dimensional dependence into bivariate copulas and yields an edge-additive objective. This enables unbiased edge-minibatched training and a variance-optimal level-wise sampling strategy that concentrates the edge budget on the most informative terms. We further reuse the sampled vine factors to build a dependence-aware particle initializer, easing subsequent SVGD refinement. Experiments on real-world tasks show consistent improvements over structured VI baselines and clear gains from efficient higher-order instance-level dependence modeling.

## 2. Background

Consider a latent-variable model $p_{\boldsymbol{\theta}}(\mathbf{X}, \mathbf{Z}) = p_{\boldsymbol{\theta}}(\mathbf{X} \mid \mathbf{Z}) \, p(\mathbf{Z})$, where $\mathbf{X} = [\mathbf{x}_1^\top, \mathbf{x}_2^\top, \cdots, \mathbf{x}_N^\top]^\top$ and $\mathbf{Z} = [\mathbf{z}_1^\top, \mathbf{z}_2^\top, \cdots, \mathbf{z}_N^\top]^\top$. Here $\mathbf{x}_i \in \mathbb{R}^{D_x}$ is the $i$-th data instance and $\mathbf{z}_i \in \mathbb{R}^D$ is its corresponding latent variable. Variational inference (VI) approximates the intractable posterior $p_{\boldsymbol{\theta}}(\mathbf{Z} \mid \mathbf{X})$ with a tractable family $q_\Phi(\mathbf{Z} \mid \mathbf{X})$ by maximizing the evidence lower bound (ELBO)

$$\mathcal{L}(\Phi, \boldsymbol{\theta}) = \mathbb{E}_{q_\Phi(\mathbf{Z}|\mathbf{X})}\big[\log p_{\boldsymbol{\theta}}(\mathbf{X}, \mathbf{Z}) - \log q_\Phi(\mathbf{Z} \mid \mathbf{X})\big]. \quad (1)$$

In the standard i.i.d. regime, one often assumes conditional independence across instances, so that $p_{\boldsymbol{\theta}}(\mathbf{X}, \mathbf{Z}) = \prod_{i=1}^N p_{\boldsymbol{\theta}}(\mathbf{x}_i \mid \mathbf{z}_i) \, p(\mathbf{z}_i)$. In this case, an instance-factorized variational family $q_{\boldsymbol{\phi}}(\mathbf{Z} \mid \mathbf{X}) = \prod_{i=1}^N q_{\boldsymbol{\phi}}(\mathbf{z}_i \mid \mathbf{x}_i)$ structurally aligns with the model and scales efficiently.

For correlated data, instance-level dependence can enter either the prior or the likelihood, and the posterior is no longer instance-factorized. A common example arises in

time series, where latents follow a Markov prior $p(\mathbf{z}_{1:N}) = p(\mathbf{z}_1) \prod_{i=2}^N p(\mathbf{z}_i \mid \mathbf{z}_{i-1})$, inducing posterior dependence across time. Instance coupling can also appear in relational or spatial data through likelihood factors on a graph, e.g., $p_{\boldsymbol{\theta}}(\mathbf{X} \mid \mathbf{Z}) = \prod_i p_{\boldsymbol{\theta}}(\mathbf{x}_i \mid \mathbf{z}_i) \prod_{(i,j) \in \mathcal{E}} p_{\boldsymbol{\theta}}(x_{ij} \mid \mathbf{z}_i, \mathbf{z}_j)$, where $x_{ij}$ is an observed interaction on edge $(i, j)$. In such settings, the mean-field form $\prod_i q(\mathbf{z}_i \mid \mathbf{x}_i)$ becomes misspecified because it cannot represent cross-instance posterior dependence.

Recent methods such as TreeVI (Xiao & Su, 2024) and HoT-VI (Xiao et al., 2025) explicitly model *instance-level* dependence in the variational posterior. TreeVI is efficient but restricts dependencies to a tree, which excludes cycles and limits the expressiveness of long-range interactions. HoT-VI introduces higher-order factors, but the cost grows rapidly with the dependency order, making it difficult to scale to long correlated batches. Moreover, tractability in these methods often relies on Gaussian parameterizations, which can be restrictive when the correlated posterior is non-Gaussian. These limitations motivate a dependence-aware posterior family with scalable learning and sampling, combined with non-parametric particle refinement to further reduce variational bias, which we develop in Section 3.

## 3. Methodology

We consider posterior inference for correlated data where the true posterior couples instance latents across $i$. Our goal is to retain this instance-level dependence while keeping inference scalable for large datasets. We first describe latent-space SVGD refinement and its projection-based amortization, then introduce a copula-structured variational posterior with scalable vine-based dependence learning and sampling.

### 3.1. Latent-Space SVGD Refinement and Projection

The instance-level structured VI methods above make posterior dependence explicit, but they can be limited by restricted dependency patterns or rapidly growing costs when richer structures are needed. An orthogonal way to improve posterior fit is to refine samples directly toward the target posterior, which leads to Stein variational gradient descent (SVGD) (Liu & Wang, 2016). SVGD transports an empirical particle distribution toward a target density by applying a deterministic transport map in an RKHS. Given a target $\pi(\mathbf{Z})$ and particles $\{\mathbf{Z}^{(k)}\}_{k=1}^M$, one SVGD step updates

$$\mathbf{Z}^{(k)} \leftarrow \mathbf{Z}^{(k)} + \epsilon \hat{\phi}(\mathbf{Z}^{(k)}), \quad (2)$$

where the empirical Stein direction is

$$\hat{\phi}(\mathbf{Z}) = \frac{1}{M} \sum_{k'=1}^M \Big[ k(\mathbf{Z}^{(k')}, \mathbf{Z}) \nabla_{\mathbf{Z}^{(k')}} \log \pi(\mathbf{Z}^{(k')}) \\ + \nabla_{\mathbf{Z}^{(k')}} k(\mathbf{Z}^{(k')}, \mathbf{Z}) \Big]. \quad (3)$$

In latent-variable models, latent-space SVGD for VAEs (Pu et al., 2017) and amortized SVGD (Feng et al., 2017) typically use SVGD as a finite-step refinement tool: an inference model generates initial particles, SVGD refines them toward the current posterior, and the refined particles are distilled back to amortize future inference. For our correlated posterior, the target is $\pi_{\boldsymbol{\theta}}(\mathbf{Z}) \propto p_{\boldsymbol{\theta}}(\mathbf{X}, \mathbf{Z})$, so Eq. (3) uses the score $\nabla_{\mathbf{z}} \log \pi_{\boldsymbol{\theta}}(\mathbf{Z}) = \nabla_{\mathbf{z}} \log p_{\boldsymbol{\theta}}(\mathbf{X}, \mathbf{Z})$.

This refinement-and-distillation procedure can be interpreted as a variational EM-like algorithm. At iteration $t$, the E-step consists of *Stein refinement* and *projection*. In the refinement stage, we draw initial particles $\{\mathbf{Z}^{(k,0)}\}_{k=1}^{M} \sim q_{\Phi^t}(\mathbf{Z} \mid \mathbf{X})$ from the current explicit variational posterior. We then apply $T$ SVGD steps targeting $\pi_{\boldsymbol{\theta}^t}(\mathbf{Z})$, yielding refined particles $\{\mathbf{Z}^{(k,T)}\}_{k=1}^{M}$. In the projection stage, the variational parameters are updated by fitting the explicit posterior family to the refined particles:

$$\Phi^{t+1} = \arg\max_{\Phi} \; \frac{1}{M} \sum_{k=1}^{M} \log q_{\Phi}\left(\mathbf{Z}^{(k,T)} \mid \mathbf{X}\right). \quad (4)$$

Given refined particles, the M-step updates model parameters by Monte Carlo maximization of the complete-data likelihood:

$$\boldsymbol{\theta}^{t+1} = \arg\max_{\boldsymbol{\theta}} \; \frac{1}{M} \sum_{k=1}^{M} \log p_{\boldsymbol{\theta}}\left(\mathbf{X}, \mathbf{Z}^{(k,T)}\right). \quad (5)$$

Our projection step follows this refinement-and-distillation spirit, but differs from Feng-style amortized SVGD in its goal. Instead of learning an unconstrained implicit sampler, we fit an explicit parametric posterior $q_{\Phi}(\mathbf{Z} \mid \mathbf{X})$, since large-scale instance-level dependence over $\mathbf{Z} = [\mathbf{z}_1^\top, \mathbf{z}_2^\top, \cdots, \mathbf{z}_N^\top]^\top$ makes fully implicit joint amortization impractical and does not provide the density required for variational learning. Thus, SVGD-refined particles are used as supervision for a tractable structured posterior, rather than as a replacement for the posterior family.

This raises a key requirement for correlated data: the projected family must itself preserve cross-instance dependence. Many latent-space SVGD pipelines use instance-wise inference models, e.g., $q_{\boldsymbol{\phi}}(\mathbf{Z} \mid \mathbf{X}) = \prod_{i=1}^{N} q_{\boldsymbol{\phi}}(\mathbf{z}_i \mid \mathbf{x}_i)$, for amortized initialization and projection. Although SVGD may temporarily induce dependence among particles, projecting them back to an instance-factorized family erases such dependence, and the next iteration again starts from independent particles. This motivates the copula-augmented posterior introduced next, whose explicit density can retain SVGD-refined cross-instance dependence while remaining trainable at the instance level.

## 3.2. Copula-Augmented Variational Posterior

To retain instance-level correlations in SVGD-based variational inference, we enlarge the amortized variational family by introducing an explicit dependence component. By Sklar's theorem (Nelsen, 2006), any multivariate distribution can be decomposed into marginal distributions and a copula term that captures their dependence. Motivated by this, we define the variational posterior for a correlated batch $\mathbf{X} = \{\mathbf{x}_i\}_{i=1}^{N}$ and latents $\mathbf{Z} = \{\mathbf{z}_i\}_{i=1}^{N}$ as

$$q_{\Phi}(\mathbf{Z} \mid \mathbf{X}) = c(\mathbf{U} \mid \mathbf{X}; \boldsymbol{\psi}) \prod_{i=1}^{N} q_{\boldsymbol{\phi}}(\mathbf{z}_i \mid \mathbf{x}_i), \quad (6)$$

where $\Phi = \{\boldsymbol{\phi}, \boldsymbol{\psi}\}$ and $c(\mathbf{U} \mid \mathbf{X}; \boldsymbol{\psi})$ is a copula *density* (i.e., a distribution on $(0, 1)^{N \times D}$) that models instance-level dependence among $\mathbf{Z}$, and $\mathbf{U}$ is obtained by applying a deterministic transformation on $\mathbf{Z}$ as elaborated subsequently. Here, $q_{\boldsymbol{\phi}}(\mathbf{z}_i \mid \mathbf{x}_i) = \int q_{\Phi}(\mathbf{Z} \mid \mathbf{X}) \, d\mathbf{Z}_{\backslash i}$ is guaranteed to hold, where $\mathbf{Z}_{\backslash i} = \{\mathbf{z}_j\}_{j \neq i}$. The copula operates on standardized variables $\mathbf{U} = [\mathbf{u}^{(1)}, \ldots, \mathbf{u}^{(D)}] \in (0, 1)^{N \times D}$. For notational convenience, we model instance-level dependence separately within each latent dimension $d = 1, \ldots, D$. For each $d$, let $\mathbf{u}^{(d)} = [u_1^{(d)}, \ldots, u_N^{(d)}]^\top$, where each $u_i^{(d)} \in (0, 1)$ is the CDF value of $z_i^{(d)}$ under the corresponding marginal posterior $q_{\boldsymbol{\phi}}(z_i^{(d)} \mid \mathbf{x}_i)$. Intuitively, we map each $z_i^{(d)}$ to the uniform space $(0, 1)$ via its CDF function and then use $c(\cdot \, ; \boldsymbol{\psi})$ to model instance-wise dependence in that space of $\mathbf{U}$. In the rest of the paper, we omit conditioning on $\mathbf{X}$ in the copula term when it is clear from context, and write $c(\mathbf{U}; \boldsymbol{\psi})$.

The copula posterior in Eq. (6) separates marginal modeling from dependence modeling. If data are i.i.d., then the copula reduces to the independence copula by simply setting $c(\mathbf{U}; \boldsymbol{\psi}) = 1$. For correlated data, the copula factor captures instance-level dependence while keeping marginal inference amortized.

We now plug Eq. (6) into the SVGD-based variational EM procedure in Section 3.1. For updating $\Phi$, we optimize the projection objective in Eq. (4). Using Eq. (6), the projected log-density decomposes as

$$\log q_{\Phi}(\mathbf{Z} \mid \mathbf{X}) = \sum_{i=1}^{N} \log q_{\boldsymbol{\phi}}(\mathbf{z}_i \mid \mathbf{x}_i) + \log c(\mathbf{U}; \boldsymbol{\psi}). \quad (7)$$

The marginal terms in Eq. (7) are straightforward because they are computed independently for each instance $i$. For efficient amortized inference, their parameters are produced by a shared marginal encoder, i.e., $[\boldsymbol{\mu}_i, \boldsymbol{\sigma}_i] = h_{\boldsymbol{\phi}}(\mathbf{x}_i)$. In contrast, the copula term $\log c(\mathbf{U}; \boldsymbol{\psi})$ is challenging because it couples the $N$ instances together through the shared dependence variables $\mathbf{U} \in (0, 1)^{N \times D}$. This leads to two practical

challenges: (1) *scalable dependence learning*, i.e., how to update $\psi$ in Eq. (4) when the dependence term couples all instances; (2) *scalable dependence-aware sampling*, i.e., how to draw initialization particles from $q_\Phi(\mathbf{Z} \mid \mathbf{X})$ while retaining instance-wise dependence encoded by $c(\mathbf{U}; \psi)$.

### 3.3. Vine-Structured Decomposition of Copula

To address the challenge of scalable dependence learning for $c(\mathbf{U}; \psi)$, one could choose a single "off-the-shelf" $N$-variate copula family (e.g., Gaussian or $t$). However, an $N$-variate copula couples all $N$ instances simultaneously, so learning and sampling must operate on the whole batch jointly, which quickly becomes impractical when $N$ is large.

Fortunately, it has been demonstrated (well known in areas such as finance and econometrics) that any absolutely continuous multivariate copula can be equivalently re-expressed as a hierarchical composition of bivariate copulas, known as the pair-copula construction (PCC) (Bedford & Cooke, 2002). In this paper, we bring the idea of vine-based pair-copula decompositions to scalable instance-level dependence modeling in variational inference.

To make the decomposition computable, PCC relies on a special hierarchical graph called a *vine*. Figure 1 shows an example vine for four variables; with the vine concept, the PCC decomposition can be stated as the following theorem.

**Theorem 3.1** (Pair-copula construction (PCC) and vine representation). *Assume $c(\mathbf{U}; \psi)$ is an absolutely continuous copula density on $\mathbf{U} \in (0,1)^{N \times D}$. Then there exists a layered edge set $E = \bigsqcup_{\ell=1}^{N-1} E_\ell$ such that*

$$c(\mathbf{U}; \psi) = \prod_{d=1}^{D} \prod_{\ell=1}^{N-1} \prod_{e \in E_\ell} c\left(u_{e,1}^{(d)}, u_{e,2}^{(d)}; \psi_{e,d}\right), \quad (8)$$

*where $c(\cdot, \cdot; \psi_{e,d})$ denotes the pair-copula term on edge $e$ (we omit any copula-family subscript for simplicity), and we apply the same construction independently for each latent dimension $d \in \{1, \ldots, D\}$. For each edge $e$, the two inputs $u_{e,1}^{(d)}, u_{e,2}^{(d)} \in (0,1)$ are obtained by the standard vine recursion: at the first level they are taken directly from $\mathbf{u}^{(d)} = (u_1^{(d)}, \ldots, u_N^{(d)})^\top$, while at higher levels they are computed deterministically from $\mathbf{u}^{(d)}$ using lower-level pair-copula terms. Detailed description is given in Appendix B.3.*

Figure 1 illustrates how a vine is built level by level. $T_1$ is a tree over the original variables, specifying which pairs are directly connected. In the example, $T_1$ contains edges $(1,2)$, $(2,3)$, and $(3,4)$, and each edge contributes a bivariate copula term such as $c(u_1, u_2)$. Here $c(u_i, u_j)$, abbreviated as $c_{i,j}$ for notation clearness, is the bivariate copula that captures the dependence between $u_i$ and $u_j$.

After $T_1$ is chosen, the second tree $T_2$ is built on top of it: each node in $T_2$ corresponds to an edge in $T_1$, and we con-

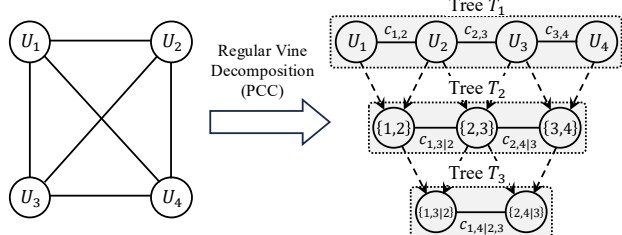

*Figure 1.* Regular vine decomposition (PCC) for four variables. Tree $T_1$ contains unconditional pair-copulas capturing dependence between $(U_1, U_2)$, $(U_2, U_3)$, and $(U_3, U_4)$, i.e., $c_{1,2}$, $c_{2,3}$, and $c_{3,4}$. Tree $T_2$ contains conditional pair-copulas that couple the "outer" variables of adjacent $T_1$ edges given their shared variable: $c_{1,3|2}$ captures dependence between $U_1$ and $U_3$ given $U_2$, and $c_{2,4|3}$ captures dependence between $U_2$ and $U_4$ given $U_3$. Tree $T_3$ adds a higher-order conditional term $c_{1,4|2,3}$, capturing the remaining dependence between $U_1$ and $U_4$ given $U_2$ and $U_3$.

nect two $T_2$ nodes only if the corresponding $T_1$ edges share a common variable. For instance, edges $(1,2)$ and $(2,3)$ share a common variable $u_2$, so $T_2$ introduces a new edge to model the dependence between the two "outer" variables $u_1$ and $u_3$ *given* the shared variable $u_2$. To exactly represent this conditional dependence (e.g., correlation between $u_1$ and $u_3$ given $u_2$), PCC shows that a deterministic transformation has to be applied to the two "outer" variables first (e.g., $u_1$ and $u_3$) and then use the bivariate copula over the transformed variables. For instance, for the leftmost edge $\{1,3 \mid 2\}$ in $T_2$, which is denoted as $e$ for simplicity, it models the conditional dependence of $u_1$ and $u_3$ given $u_2$. For notation simplicity, we denote the transformed variables of $u_1$ and $u_3$ as $u_{e,1}$ and $u_{e,2}$, respectively. Thus, the conditional dependence between $u_1$ and $u_3$ given $u_2$ can be written as a bivariate copula term $c(u_{e,1}, u_{e,2})$ (corresponding to $c_{1,3|2}$ in Figure 1); similarly, another $T_2$ edge yields the term corresponding to $c_{2,4|3}$.

Once the dependence captured by $T_1$ and $T_2$ is taken into account, there may still be residual correlation between the endpoints of the chain, i.e., the dependence between $u_1$ and $u_4$ given $(u_2, u_3)$. The vine recursion therefore computes two conditional pseudo-observations for the $T_3$ edge, denoted by $u_{e,1} = F(u_1 \mid u_2, u_3)$ and $u_{e,2} = F(u_4 \mid u_2, u_3)$, and models their remaining dependence with a bivariate term $c(u_{e,1}, u_{e,2})$, corresponding to $c_{1,4|2,3}$. This layer-by-layer construction generalizes to arbitrary $N$, yielding a sequence of trees where each layer adds bivariate copulas that capture higher-order dependence omitted in lower trees.

Finally, taking the logarithm of the PCC factorization in Eq. (8) gives an edge-additive decomposition:

$$\log c(\mathbf{U}; \psi) = \sum_{d=1}^{D} \sum_{\ell=1}^{N-1} \sum_{e \in E_\ell} \log c\left(u_{e,1}^{(d)}, u_{e,2}^{(d)}; \psi_{e,d}\right).$$
$$(9)$$

This sum structure is the key for us to achieve scalability, as shown in our subsequent sections.

**Pair-copula parameterization.** For scalability, we do not assign an independent free parameter $\psi_{e,d}$ to every pair-copula factor. Instead, for each vine edge $e$, we use a neural network to directly output its copula parameter. Let $C(e)$ denote the two conditioned variables of edge $e$ with $|C(e)| = 2$, and let $D(e)$ denote its conditioning set; for example, for $e = \{1, 4 \mid 2, 3\}$, $C(e) = \{1, 4\}$ and $D(e) = \{2, 3\}$. We parameterize

$$\psi_e = f_{\boldsymbol{\eta}}\big(\{[\boldsymbol{\mu}_i, \boldsymbol{\sigma}_i]\}_{i \in C(e)}, \mathrm{Pool}\big(\{[\boldsymbol{\mu}_j, \boldsymbol{\sigma}_j]\}_{j \in D(e)}\big)\big), \tag{10}$$

where $f_{\boldsymbol{\eta}}(\cdot)$ is a shared lightweight neural network with parameter $\boldsymbol{\eta}$; $\boldsymbol{\mu}_i$ and $\boldsymbol{\sigma}_i$ are the mean and standard deviation of the amortized marginal posterior $q_{\boldsymbol{\phi}}(\mathbf{z}_i \mid \mathbf{x}_i)$, respectively; and $\mathrm{Pool}(\cdot)$ is a simple aggregation over the conditioning variables, like averaging operation. This parameterization allows copula parameters to depend on local marginal and conditional contexts while avoiding a fully independent parameterization over all vine edges. More implementation details are provided in Appendix C.4.

### 3.4. Scalable Vine Learning via Level-Wise Sampling

Eq. (9) decomposes the copula log-density into a sum of pair-copula edge terms, but a full vine contains $\mathcal{O}(N^2)$ edges. Computing all edge terms and their gradients at every update in Eq. (4) is prohibitive for large $N$. We address this by subsampling vine edges to obtain an unbiased gradient estimator, and propose a method to reduce its variance by allocating the edge budget across vine levels.

#### 3.4.1. EDGE-MINIBATCHED UNBIASED GRADIENTS

Define the per-edge, per-dimension gradient contribution

$$\mathbf{g}_{e,d}(\boldsymbol{\psi}; \mathbf{U}) = \nabla_{\boldsymbol{\psi}} \log c_{e,d}\big(u_{e,1}^{(d)}, u_{e,2}^{(d)}; \boldsymbol{\psi}_{e,d}\big). \tag{11}$$

From Eq. (9), the full gradient of the copula term can be written as

$$\nabla_{\boldsymbol{\psi}} \log c(\mathbf{U}; \boldsymbol{\psi}) = \sum_{d=1}^{D} \sum_{\ell=1}^{N-1} \sum_{e \in E_\ell} \mathbf{g}_{e,d}(\boldsymbol{\psi}; \mathbf{U}). \tag{12}$$

Let $S \subset E$ be a random minibatch of edges sampled at each iteration, and let $\pi_e \in (0, 1]$ denote the inclusion probability of edge $e$, i.e., how often $e$ is expected to appear in $S$. We use Horvitz–Thompson reweighting (Horvitz & Thompson, 1952): edges sampled less frequently (smaller $\pi_e$) are given larger weights $1/\pi_e$ so that their contributions are properly accounted for on average.

$$\widehat{\nabla}_{\boldsymbol{\psi}} \log c(\mathbf{U}; \boldsymbol{\psi}) = \sum_{d=1}^{D} \sum_{e \in S} \frac{1}{\pi_e} \mathbf{g}_{e,d}(\boldsymbol{\psi}; \mathbf{U}). \tag{13}$$

This reweighting yields an unbiased estimator of the full edge-sum gradient, formalized in the following theorem.

**Theorem 3.2** (Unbiased edge-minibatch estimator). *Assume* $\pi_e > 0$ *for all* $e \in E$. *Then*

$$\mathbb{E}_S\Big[\widehat{\nabla}_{\boldsymbol{\psi}} \log c(\mathbf{U}; \boldsymbol{\psi})\Big] = \nabla_{\boldsymbol{\psi}} \log c(\mathbf{U}; \boldsymbol{\psi}). \tag{14}$$

Theorem 3.2 guarantees that optimizing the projection objective in Eq. (4) with edge minibatches remains faithful to the full-vine objective in expectation.

#### 3.4.2. VARIANCE-OPTIMAL LEVEL-WISE SAMPLING

Uniform edge sampling is unbiased but can have high variance, because different vine levels may contribute gradients of very different magnitudes. We therefore sample edges *level by level*: for each level $\ell \in \{1, \ldots, N-1\}$, we sample approximately $B_\ell$ edges from $E_\ell$ and set the sampled subset of edges $S = \bigsqcup_{\ell=1}^{N-1} S_\ell$ as the union of edges sampled from all levels. If sampling is uniform within level $\ell$, then each edge $e \in E_\ell$ has inclusion probability approximately $\pi_e \approx B_\ell / |E_\ell|$. The remaining question is how to choose $\{B_\ell\}$ so that the resulting Horvitz–Thompson gradient estimator has low variance. To this end, we quantify how much gradient signal each level contributes via the following level-wise gradient energy:

$$\Omega_\ell = \sum_{d=1}^{D} \sum_{e \in E_\ell} \mathbb{E}\Big[\|\mathbf{g}_{e,d}(\boldsymbol{\psi}; \mathbf{U})\|_2^2\Big], \tag{15}$$

which measures how much gradient signal level $\ell$ contributes on average; allocating samples to high-energy levels reduces the variance of the estimator under a fixed budget.

**Theorem 3.3** (Variance-optimal stratified allocation). *Consider Horvitz–Thompson estimation of* $\nabla_{\boldsymbol{\psi}} \log c(\mathbf{U}; \boldsymbol{\psi})$ *under stratified sampling across levels* $\{E_\ell\}$ *with fixed total expected budget* $B$. *Under the standard stratified variance approximation that treats within-level samples as independent and uses the energies* $\{\Omega_\ell\}$ *in Eq. (15), the allocation that minimizes the estimator variance is*

$$B_\ell^* = B \cdot \frac{\Omega_\ell}{\sum_{j=1}^{L} \Omega_j}. \tag{16}$$

Theorem 3.3 yields a simple rule: sample more edges from vine levels with larger expected gradient energy. Combined with Theorem 3.2, this gives an estimator that is both correct (unbiased) and efficient (variance-controlled) for updating $\boldsymbol{\psi}$ in the projection step. Figure 3 in Appendix A.1 intuitively illustrates why structure-aware level-wise sampling better aligns the sampling probability with gradient energy than uniform sampling under a fixed budget.

### 3.5. Efficient Sampling from Vine with Sampled Edges

As pointed out in Section 3.2, each E-step needs initial particles from $q_\Phi(\mathbf{Z} \mid \mathbf{X})$ for subsequent SVGD updating, but sampling from a full vine costs $O(N^2)$ per sample is expensive. In each iteration, Section 3.4 only models and updates the dependence carried by a subset of edges $S \subset E$. We therefore propose to reuse the same $S$ to define a sparse vine to draw initialization particles.

Formally, define the sparse variational posterior

$$q_{\Phi,S}(\mathbf{Z} \mid \mathbf{X}) = c_S(\mathbf{U}; \boldsymbol{\psi}) \prod_{i=1}^{N} q_{\boldsymbol{\phi}}(\mathbf{z}_i \mid \mathbf{x}_i), \quad (17)$$

where $c_S(\mathbf{U}; \boldsymbol{\psi})$ denotes the vine copula obtained by keeping only the pair-copula factors associated with edges in $S$ and setting all other pair-copula factors to the independence copula. We sample particles from $q_{\Phi,S}$ via an inverse Rosenblatt transform on this sparse vine, and then apply SVGD refinement using the exact posterior score.

In our training loop, we update the copula parameters using only a sampled edge set $S_t \subset E$ at iteration $t$ (Section 3.4). To keep sampling consistent with what is being learned in that iteration, we reuse the same $S_t$ to define the sparse initializer $q_{\Phi^t, S_t}$. A natural concern is whether initializing from $q_{\Phi^t, S_t}$ introduces a bias that persists after refinement. The following theorem shows that this initialization bias is transient: as the refinement budget increases, the refined distributions become asymptotically insensitive to whether we start from the sparse initializer or the full $q_{\Phi^t}$.

**Theorem 3.4** (Dynamic convergence of sparse-init refinement). *Consider the iterative updates of $\Phi$ using edge minibatches $S_t$ with $\pi_e > 0$ for all $e \in E$. Under standard stochastic approximation conditions, $\Phi^t$ converges to a stationary point of the* full-vine *projection objective. Moreover, with a refinement schedule $T_t \to \infty$, the SVGD-refined distributions initialized from $q_{\Phi^t, S_t}$ and from $q_{\Phi^t}$ satisfy*

$$d_{\mathrm{TV}}\left( \mathcal{K}_{\boldsymbol{\theta}}^{T_t}(q_{\Phi^t, S_t}), \ \mathcal{K}_{\boldsymbol{\theta}}^{T_t}(q_{\Phi^t}) \right) \ \xrightarrow[t\to\infty]{\mathbb{P}} \ 0.$$

*A complete statement with assumptions and a detailed proof are provided in Appendix B.7.1.*

Theorem 3.4 shows that sparse-vine sampling introduces only a transient initialization bias: as $t$ grows and the refinement budget increases, the refined distributions starting from $q_{\Phi^t, S_t}$ and $q_{\Phi^t}$ become asymptotically indistinguishable. In practice, we use a finite number of SVGD steps, so it is also important to understand which omitted edges dominate the remaining gap; we characterize this next via omitted-edge energies.

**Theorem 3.5** (KL gap is governed by omitted-edge energies). *Under standard regularity conditions and a second-order expansion around independence for omitted edges,*

*the sparse-vine approximation satisfies*

$$D_{\mathrm{KL}}[q_\Phi(\mathbf{Z} \mid \mathbf{X}) \,\|\, q_{\Phi,S}(\mathbf{Z} \mid \mathbf{X})] \approx \frac{1}{2} \sum_{\ell=1}^{L} \sum_{e \in E_\ell \setminus S} \omega_e. \quad (18)$$

*Here $\omega_e = \sum_{d=1}^{D} \boldsymbol{\psi}_{e,d}^\top \mathbf{I}_{e,d} \boldsymbol{\psi}_{e,d} \geq 0$ denotes the omitted-edge energy of edge $e$, where $\mathbf{I}_{e,d}$ is the Fisher information of the pair-copula family at the independence copula for latent dimension $d$ (see Appendix B.7 for derivation).*

Theorem 3.5 characterizes the approximation gap via omitted-edge energies. Edges with larger $\omega_e$ contribute more to the KL gap and therefore should be prioritized in sparse initialization and learning. Together, the two theorems provide a coherent rationale for reusing the same minibatch $S$ across projection and sampling: level-wise, energy-aware edge sampling simultaneously (i) reduces gradient variance for learning $c(\mathbf{U}; \boldsymbol{\psi})$ and (ii) concentrates sparse initialization on the most important edges, making SVGD both cheaper and more effective.

## 4. Related Work

Variational inference often trades expressiveness for tractability, with mean-field assumptions remaining a default for scalability. To increase flexibility, *dimension-level* structured variational families enrich the posterior within each latent vector, e.g., full-covariance Gaussians (Opper & Archambeau, 2009; Titsias & Lázaro-Gredilla, 2014), normalizing flows (Rezende & Mohamed, 2015; Papamakarios et al., 2021), implicit posteriors (Huszár, 2017; Shi et al., 2018), hierarchical constructions (Ranganath et al., 2016; Tran et al., 2015), and mixture-based families (Dilokthanakul et al., 2016; Tomczak & Welling, 2018). While effective for correlations across latent *dimensions*, these approaches do not address *instance-level* coupling induced by correlated data, where correlated segments can be long and generic dense instance dependence becomes impractical.

**Structured VI.** A smaller line of work targets *instance-level* structured inference by coupling latents across instances. CVAE-style approaches (Tang et al., 2019; Ou et al., 2021) and TreeVI (Xiao & Su, 2024) build tree-structured posteriors, which scale well but are limited to first-order interactions along an acyclic backbone. HoT-VI (Xiao et al., 2025) increases expressiveness with higher-order dependencies, but its cost grows quickly with the order, limiting scalability to long-range or high-order correlations; moreover, such methods often rely on Gaussian or restricted conditional forms for tractability. Copula-SVI avoids these restrictions by decoupling scalable amortized marginal inference from instance-level dependence modeling, and represents complex cross-instance dependence with a regular vine of tractable bivariate pair-copulas.

**Copula-based VI.** Copula-based variational inference has also been explored to enrich posterior dependence beyond mean-field assumptions. Tran et al. (2015) introduced a copula variational family and discussed vine copulas as a scalable construction; later works developed Gaussian, implicit, high-dimensional, or block-dependent copula approximations (Han et al., 2016; Smith et al., 2020; Smith & Loaiza-Maya, 2023; Fu et al., 2025). However, extending these reparameterized copula-VI schemes to large-scale instance-level dependence is nontrivial. Full inverse-Rosenblatt sampling requires dense recursive conditional transformations, while sparse or truncated vines completely change the sampling distribution and the lower bound objective. Moreover, since reparameterized learning differentiates through the sampling path, gradients must pass through these nested inversions, making deep instance-level vines computationally and numerically burdensome. Copula-SVI decouples sampling from full-vine learning: sparse-vine samples are used only as dependence-aware initialization, SVGD then corrects the particles toward the target posterior, and the explicit full-vine posterior is then fitted to the corrected samples with the unbiased edge-minibatched gradients. Compared with prior copula-based VI, the key distinction is this scalable training mechanism for full-vine instance-level dependence, rather than the copula family alone.

**Stein-based VI.** Stein variational gradient descent (SVGD) (Liu & Wang, 2016; Liu, 2017) transports particles toward a target posterior through functional gradients. Latent-space and amortized SVGD methods (Pu et al., 2017; Feng et al., 2017) refine particles and then train a reusable inference model from the refined particles. Our method follows this refinement-and-projection view, but fits an explicit vine-copula posterior rather than an implicit sampler. While implicit amortized samplers are flexible for within-instance latent dimensions, modeling cross-instance dependence would require a joint network over all correlated instances, which is impractical for large $N$. By restricting the posterior to a parametric vine-copula family, Copula-SVI decomposes instance-level dependence into bivariate factors, so the dependence network avoids taking all $N$ instances as joint input, and instead processes only the two endpoints of each vine edge while still preserving correlations over all instances.

## 5. Experiments

We evaluate **Copula-SVI** on three tasks: time series anomaly detection, time series forecasting, and constrained clustering. For anomaly detection, we compare against DAGMM, LSTM-VAE, OmniAnomaly, SISVAE, and HoT-VI. For forecasting, we compare against Informer, GRU-NVP, DeepAR, VRAE, and HoT-VI. For constrained clustering, we compare against PCKMeans, SDEC, C-IDEC,

DC-GMM, VaDE, DGG, TreeVI, and HoT-VI. See Appendix C for details. Code is available at: https://github.com/Mephestopheles/Copula-SVI.

### 5.1. Constrained Clustering

Constrained clustering injects instance-level supervision by specifying which pairs of data should share a cluster (*must-link*) or should not (*cannot-link*). We encode constraints with a graph $\mathcal{G} = (\mathcal{V}, \mathcal{E}, \mathbf{A})$ over $N$ instances, where $\mathbf{A} \in \mathbb{R}^{N \times N}$ stores both type and confidence: $[\mathbf{A}]_{ij} > 0$ for must-link, $[\mathbf{A}]_{ij} < 0$ for cannot-link, and $[\mathbf{A}]_{ij} = 0$ otherwise, with $|[\mathbf{A}]_{ij}|$ indicating strength. We follow a probabilistic formulation with observed data $\mathbf{X} = \{\mathbf{x}_i\}_{i=1}^N$, latent embeddings $\mathbf{Z} = \{\mathbf{z}_i\}_{i=1}^N$, and cluster assignments $\mathbf{c} = \{c_i\}_{i=1}^N$, where the complete-data distribution factorizes as $p_{\boldsymbol{\theta}}(\mathbf{X}, \mathbf{Z}, \mathbf{c} \mid \mathbf{A}) = p_{\boldsymbol{\theta}}(\mathbf{X} \mid \mathbf{Z}) \, p(\mathbf{Z} \mid \mathbf{c}) \, p(\mathbf{c} \mid \mathbf{A})$. The constraint graph enters through the weighted assignment prior $p(\mathbf{c} \mid \mathbf{A}) = \frac{1}{\Omega(\boldsymbol{\pi})} \prod_i \pi_{c_i} \prod_{j \neq i} \exp([\mathbf{A}]_{ij} \, \mathbb{I}[c_i = c_j])$, which couples $\mathbf{c}$ across instances and induces instance-level posterior dependence among $\mathbf{Z}$. We defer the detailed specification of $p_{\boldsymbol{\theta}}(\mathbf{X} \mid \mathbf{Z})$ and $p(\mathbf{Z} \mid \mathbf{c})$ to Appendix C.

For inference, we use $q_{\Phi}(\mathbf{Z}, \mathbf{c} \mid \mathbf{X}) = q_{\Phi}(\mathbf{Z} \mid \mathbf{X}) \, q(\mathbf{c} \mid \mathbf{Z})$, where $q(\mathbf{c} \mid \mathbf{Z})$ is computed via Bayes' rule under $p(\mathbf{Z} \mid \mathbf{c})p(\mathbf{c} \mid \mathbf{A})$. The core distinction among methods is the expressiveness of $q_{\Phi}(\mathbf{Z} \mid \mathbf{X})$: DC-GMM uses an instance-factorized encoder and thus ignores constraint-induced dependence; TreeVI introduces tree-structured instance dependence and captures only first-order interactions; HoT-VI extends TreeVI with $k$-order dependencies but its cost grows quickly with $k$. In contrast, **Copula-SVI** uses a copula-structured posterior to explicitly model instance-level dependence and then applies SVGD refinement in the joint latent space to non-parametrically reduce residual variational bias. We use a $k$-truncated regular vine with $k \in \{5, 10, 50, 100\}$; the first-tree structure is constructed from $\mathcal{G}$ in the same spirit as TreeVI, and each pair-copula is instantiated as a Gaussian copula.

**Quantitative results.** We report ACC, NMI, and ARI averaged over 10 runs. Table 1 shows that Copula-SVI consistently outperforms DC-GMM and TreeVI, demonstrating the benefit of modeling instance-level posterior dependence beyond i.i.d. factorization (DC-GMM) and beyond first-order tree structure (TreeVI). Copula-SVI outperforms HoT-VI at equal dependence orders, indicating that SVGD refinement reduces residual bias beyond explicit modeling. Performance improves with $k$, suggesting richer dependence enables effective constraint propagation.

**Runtime comparison & Visualization.** Figure 2a shows runtime on MNIST versus dependence order. Copula-SVI remains efficient as $k$ increases, with runtime comparable to DC-GMM and TreeVI, while scaling substantially better

*Table 1.* Clustering performances (%) of Copula-SVI compared with baselines. Means and standard deviations are computed across 10 runs with different random initializations. † Results are taken from TreeVI (Xiao & Su, 2024).

| Dataset | Metric | VaDE | C-IDEC | DGG† | DC-GMM | TreeVI | HoT-VI | | Copula-SVI *(Ours)* | | | |
|---|---|---|---|---|---|---|---|---|---|---|---|---|
| | | | | | | | 5-order | 10-order | 5-order | 10-order | 50-order | 100-order |
| MNIST | ACC | 89.0±5.0 | 96.3±0.2 | 95.8±0.1 | 96.5±0.2 | 97.4±0.3 | 98.3±0.4 | 98.5±0.3 | 98.3±0.3 | 98.6±0.2 | 98.8±0.3 | **99.0±0.3** |
| | NMI | 82.8±3.0 | 91.8±1.0 | 91.2±0.2 | 91.4±0.3 | 93.1±0.6 | 94.2±0.3 | 94.6±0.3 | 95.6±0.4 | 96.2±0.3 | 96.4±0.3 | **96.5±0.4** |
| | ARI | 80.9±3.0 | 92.1±0.4 | 91.4±0.3 | 92.5±0.5 | 93.7±0.7 | 95.2±0.5 | 95.6±0.5 | 96.4±0.2 | 96.7±0.1 | 96.9±0.4 | **97.0±0.2** |
| fMNIST | ACC | 55.1±2.2 | 68.1±3.0 | 79.9±0.4 | 80.5±0.8 | 81.4±0.6 | 83.2±0.5 | 83.4±0.4 | 83.6±0.3 | 83.8±0.3 | 84.2±0.4 | **84.4±0.3** |
| | NMI | 57.9±2.7 | 66.7±2.0 | 70.1±0.3 | 72.0±0.4 | 73.9±0.6 | 74.8±0.5 | 75.1±0.4 | 75.2±0.3 | 75.6±0.2 | 75.9±0.2 | **76.1±0.3** |
| | ARI | 41.6±3.1 | 52.3±3.0 | 64.9±0.3 | 66.4±0.5 | 67.9±0.9 | 69.1±0.5 | 69.2±0.4 | 69.4±0.1 | 69.5±0.2 | 69.9±0.3 | **70.0±0.2** |
| Reuters | ACC | 76.0±0.7 | 94.7±0.6 | 93.5±0.6 | 95.4±0.2 | 95.9±0.6 | 97.2±0.6 | 97.6±0.5 | 97.5±0.2 | 98.0±0.4 | 98.2±0.3 | **98.4±0.2** |
| | NMI | 50.1±1.3 | 81.4±0.7 | 81.2±0.8 | 82.7±0.7 | 83.4±0.5 | 85.1±0.6 | 85.4±0.5 | 85.6±0.2 | 85.8±0.2 | 86.3±0.2 | **86.4±0.5** |
| | ARI | 58.0±1.4 | 87.7±0.9 | 87.8±0.5 | 89.0±0.6 | 90.2±0.4 | 91.6±0.5 | 92.0±0.4 | 91.8±0.3 | 92.5±0.3 | 92.6±0.4 | **92.8±0.3** |
| STL-10 | ACC | 77.3±0.5 | 81.6±3.8 | 89.9±0.3 | 89.5±0.5 | 90.4±0.9 | 92.2±0.6 | 92.4±0.4 | 92.7±0.2 | 92.8±0.2 | 93.0±0.3 | **93.3±0.3** |
| | NMI | 70.6±0.4 | 77.3±1.7 | 80.9±0.5 | 80.2±0.7 | 81.3±0.8 | 82.8±0.6 | 83.1±0.5 | 83.0±0.3 | 83.4±0.3 | 83.8±0.3 | **84.0±0.2** |
| | ARI | 62.7±0.4 | 71.8±3.4 | 79.0±0.4 | 78.4±0.9 | 79.5±0.7 | 81.3±0.5 | 81.5±0.5 | 81.5±0.3 | 81.9±0.2 | 82.2±0.3 | **82.3±0.4** |

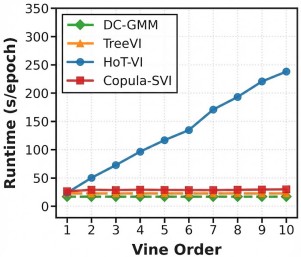

*(a)* Runtime comparison (per epoch) vs. dependence order.

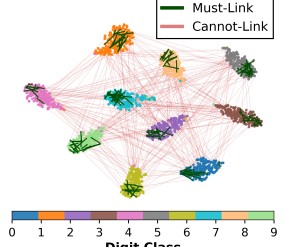

*(b)* t-SNE visualization of latent space with constraints.

*Figure 2.* Constrained clustering results on MNIST dataset.

*Table 2.* Anomaly detection results (F1-score and ELBO).

| Dataset | | SMAP | | MSL | | SMD | |
|---|---|---|---|---|---|---|---|
| Metric | | F1 | ELBO | F1 | ELBO | F1 | ELBO |
| DAGMM | | 0.7105 | -115.2820 | 0.7007 | -277.7380 | 0.7094 | -155.9460 |
| LSTM-VAE | | 0.7298 | -116.9500 | 0.6780 | -281.3220 | 0.7842 | -146.0540 |
| OmniAnomaly | | 0.8434 | -98.9217 | 0.8849 | -161.0002 | 0.8857 | -72.0419 |
| SISVAE | | 0.8299 | -101.1878 | 0.8766 | -182.6060 | 0.8775 | -72.5832 |
| HoT-VI | 3-order | 0.8552 | -95.2314 | 0.8940 | -157.2134 | 0.9153 | -67.4001 |
| | 10-order | 0.8636 | -92.2948 | 0.9145 | -134.0815 | 0.9284 | -65.0345 |
| *Ours* | 3-order | 0.8628 | -93.9194 | 0.9032 | -154.5485 | 0.9224 | -65.1764 |
| | 10-order | 0.8753 | -90.9828 | 0.9245 | -132.6568 | 0.9371 | -63.7555 |
| | 50-order | 0.8846 | -87.4992 | 0.9387 | -103.7108 | 0.9474 | -62.0640 |
| | 100-order | **0.8871** | **-85.0005** | **0.9453** | **-91.1058** | **0.9504** | **-61.1717** |

than HoT-VI at larger orders. Figure 2b visualizes MNIST embeddings with constraints at $k = 10$. The clusters are tighter and better separated, and constraint edges align with the intended effect of $\mathbf{A}$. This improvement is consistent with modeling high-order posteriors: dependence allows constraints to influence not only constrained pairs but also nearby points through the learned dependence structure, yielding a more globally coherent clustering geometry.

## 5.2. Time Series Modeling

Real-world time series are not composed of independent segments: neighboring windows often share regimes, sea-

sonality, and latent causes. After segmenting a sequence into $\mathbf{X} = \{\mathbf{x}_i\}_{i=1}^N$ with $\mathbf{x}_i \in \mathbb{R}^{T \times C}$, this induces *instance-level* dependence among the window latents $\{\mathbf{z}_i\}_{i=1}^N$. However, many VAE-based approaches still perform inference with an instance-factorized posterior $q(\mathbf{Z} \mid \mathbf{X}) = \prod_{i=1}^N q(\mathbf{z}_i \mid \mathbf{x}_i)$, treating windows as i.i.d. during posterior inference, which can weaken temporal coherence and limit evidence propagation across windows. We address this mismatch by applying Copula-SVI *across windows*: we use a window-factorized decoder $p_{\boldsymbol{\theta}}(\mathbf{X} \mid \mathbf{Z}) = \prod_{i=1}^N p_{\boldsymbol{\theta}}(\mathbf{x}_i \mid \mathbf{z}_i)$ (GRU-based to capture within-window dynamics), while modeling cross-window dependence through the copula-structured variational posterior $q_{\Phi}(\mathbf{Z} \mid \mathbf{X}) = c(\mathbf{U}; \boldsymbol{\psi}) \prod_{i=1}^N q_{\boldsymbol{\phi}}(\mathbf{z}_i \mid \mathbf{x}_i)$, followed by joint SVGD refinement in the latent space. We instantiate $c(\mathbf{U}; \boldsymbol{\psi})$ as a $k$-truncated regular vine with Gaussian pair-copulas and vary $k$ to study the effect of higher-order cross-window dependence; full dataset and training details are given in Appendix C.

**Time Series Anomaly Detection.** The objective is to detect anomalous windows by training on normal data and flagging test windows with low model fit. After segmenting each sequence into $\{\mathbf{x}_i\}$, we train on normal windows and compute an anomaly score per window using reconstruction-based likelihood (equivalently, negative ELBO). Modeling $q_{\Phi}(\mathbf{Z} \mid \mathbf{X})$ with cross-window dependence is important because anomalies often manifest as deviations from a temporally coherent context shared by nearby windows, and an i.i.d. posterior may fail to leverage this context. We evaluate performance using F1-score and report ELBO as a complementary modeling metric. Table 2 shows that Copula-SVI achieves better F1 and ELBO than baselines, and performance improves consistently as $k$ increases, indicating that richer cross-window dependence helps propagate contextual evidence. Compared with HoT-VI, Copula-SVI attains stronger results while remaining practical at larger orders due to the truncated-vine parameterization and the addi-

*Table 3.* Multivariate time series forecasting results (MSE/MAE) across horizons $H \in \{24, 48, 168, 336, 720\}$. Best performance is highlighted in bold font and the second best results are underlined. † Results are taken from HoT-VI (Xiao et al., 2025).

| Method | | Informer† MSE | MAE | GRU-NVP† MSE | MAE | DeepAR† MSE | MAE | VRAE† MSE | MAE | HoT-VI 3-order MSE | MAE | 10-order MSE | MAE | Copula-SVI *(Ours)* 3-order MSE | MAE | 10-order MSE | MAE | 50-order MSE | MAE | 100-order MSE | MAE |
|---|---|---|---|---|---|---|---|---|---|---|---|---|---|---|---|---|---|---|---|---|---|
| ETTh1 | 24 | 0.577 | 0.549 | 3.540 | 0.733 | 1.166 | 0.836 | 0.743 | 0.762 | 0.543 | 0.505 | 0.363 | 0.376 | 0.521 | 0.472 | 0.344 | 0.349 | 0.330 | 0.337 | **0.321** | **0.326** |
| | 48 | 0.685 | 0.625 | 2.549 | 0.622 | 1.154 | 0.827 | 0.826 | 0.801 | 0.578 | 0.528 | 0.392 | 0.392 | 0.545 | 0.487 | 0.370 | 0.373 | 0.348 | 0.354 | **0.341** | **0.350** |
| | 168 | 0.931 | 0.752 | 3.831 | 0.774 | 1.083 | 0.778 | 1.070 | 0.938 | 0.721 | 0.615 | 0.510 | 0.464 | 0.684 | 0.568 | 0.489 | 0.442 | 0.472 | 0.417 | **0.462** | **0.413** |
| | 336 | 1.128 | 0.873 | 6.877 | 1.008 | 1.043 | 0.766 | 1.199 | 1.016 | 0.883 | 0.702 | 0.616 | 0.525 | 0.817 | 0.671 | 0.575 | 0.494 | 0.543 | 0.465 | **0.537** | **0.459** |
| | 720 | 1.215 | 0.896 | 5.377 | 1.060 | 1.075 | 0.795 | 1.426 | 1.164 | 1.021 | 0.781 | 0.763 | 0.630 | 0.959 | 0.726 | 0.704 | 0.582 | 0.663 | 0.551 | **0.643** | **0.537** |
| ETTm1 | 24 | 0.453 | 0.444 | 0.605 | 0.437 | 1.360 | 0.871 | 0.687 | 0.646 | 0.409 | 0.417 | 0.253 | 0.298 | 0.385 | 0.387 | 0.242 | 0.278 | 0.233 | 0.270 | **0.226** | **0.263** |
| | 48 | 0.494 | 0.503 | 2.787 | 0.701 | 1.334 | 0.866 | 0.817 | 0.724 | 0.535 | 0.488 | 0.330 | 0.345 | 0.497 | 0.453 | 0.310 | 0.325 | 0.294 | 0.315 | **0.287** | **0.306** |
| | 168 | 0.678 | 0.614 | 4.212 | 0.824 | 1.170 | 0.838 | 0.853 | 0.794 | 0.578 | 0.521 | 0.368 | 0.373 | 0.547 | 0.492 | 0.349 | 0.346 | 0.333 | 0.333 | **0.323** | **0.324** |
| | 336 | 1.056 | 0.786 | 5.062 | 1.019 | 1.249 | 0.846 | 1.091 | 0.975 | 0.641 | 0.567 | 0.434 | 0.415 | 0.605 | 0.526 | 0.410 | 0.396 | 0.386 | 0.382 | **0.377** | **0.369** |
| | 720 | 1.192 | 0.926 | 5.799 | 1.075 | 1.075 | 0.770 | 1.165 | 0.996 | 0.737 | 0.626 | 0.528 | 0.474 | 0.686 | 0.583 | 0.501 | 0.441 | 0.475 | 0.427 | **0.462** | **0.419** |
| Electricity | 24 | 0.312 | 0.387 | 3.514 | 1.844 | 0.211 | 0.330 | 0.279 | 0.396 | 0.256 | 0.346 | 0.134 | 0.238 | 0.240 | 0.330 | 0.124 | 0.222 | 0.120 | 0.209 | **0.117** | **0.206** |
| | 48 | 0.392 | 0.431 | 3.318 | 1.786 | 0.332 | 0.398 | 0.317 | 0.410 | 0.277 | 0.363 | 0.152 | 0.255 | 0.266 | 0.347 | 0.140 | 0.244 | 0.135 | 0.233 | **0.133** | **0.229** |
| | 168 | 0.515 | 0.509 | 3.482 | 1.833 | 1.065 | 0.811 | 0.366 | 0.475 | 0.303 | 0.382 | 0.174 | 0.273 | 0.283 | 0.363 | 0.162 | 0.255 | 0.154 | 0.240 | **0.150** | **0.236** |
| | 336 | 0.759 | 0.625 | 3.921 | 1.941 | 1.040 | 0.795 | 0.402 | 0.515 | 0.319 | 0.395 | 0.194 | 0.293 | 0.301 | 0.363 | 0.183 | 0.281 | 0.175 | 0.271 | **0.172** | **0.266** |
| | 720 | 0.969 | 0.788 | 4.232 | 2.020 | 1.048 | 0.804 | 0.450 | 0.556 | 0.348 | 0.416 | 0.230 | 0.323 | 0.331 | 0.383 | 0.217 | 0.299 | 0.207 | 0.281 | **0.204** | **0.273** |
| Exchange | 24 | 0.611 | 0.626 | 1.557 | 0.877 | 1.328 | 0.692 | 0.140 | 0.310 | 0.093 | 0.227 | 0.033 | 0.126 | 0.053 | 0.145 | 0.028 | 0.108 | 0.026 | 0.101 | **0.025** | **0.097** |
| | 48 | 0.680 | 0.644 | 1.589 | 0.883 | 1.345 | 0.701 | 0.238 | 0.435 | 0.171 | 0.306 | 0.058 | 0.164 | 0.082 | 0.163 | 0.049 | 0.141 | 0.046 | 0.134 | **0.043** | **0.128** |
| | 168 | 1.097 | 0.825 | 1.663 | 0.903 | 1.434 | 0.745 | 0.642 | 0.703 | 0.368 | 0.458 | 0.196 | 0.326 | 0.226 | 0.311 | 0.165 | 0.275 | 0.155 | 0.254 | **0.150** | **0.242** |
| | 336 | 1.672 | 1.036 | 1.682 | 0.905 | 1.489 | 0.778 | 1.050 | 0.953 | 1.165 | 0.821 | 0.496 | 0.515 | 0.576 | 0.556 | 0.410 | 0.432 | 0.385 | 0.408 | **0.367** | **0.390** |
| | 720 | 2.478 | 1.310 | 1.748 | 0.928 | 1.526 | 0.793 | 3.003 | 1.593 | 2.029 | 1.090 | 1.508 | 0.857 | 1.368 | 0.823 | 1.247 | 0.744 | 1.180 | 0.703 | **1.139** | **0.681** |
| Weather | 24 | 0.162 | 0.235 | 1.222 | 0.909 | 0.205 | 0.250 | 0.227 | 0.315 | 0.186 | 0.281 | 0.129 | 0.179 | 0.174 | 0.262 | 0.123 | 0.167 | 0.117 | 0.158 | **0.116** | **0.156** |
| | 48 | 0.348 | 0.400 | 2.319 | 1.287 | 0.229 | 0.267 | 0.449 | 0.495 | 0.291 | 0.361 | 0.186 | 0.230 | 0.271 | 0.345 | 0.174 | 0.215 | 0.166 | 0.203 | **0.164** | **0.197** |
| | 168 | 0.444 | 0.463 | 2.174 | 1.165 | 0.344 | 0.343 | 0.563 | 0.648 | 0.429 | 0.486 | 0.294 | 0.313 | 0.406 | 0.453 | 0.271 | 0.289 | 0.256 | 0.277 | **0.253** | **0.268** |
| | 336 | 0.578 | 0.523 | 2.119 | 1.221 | 0.568 | 0.527 | 0.781 | 0.841 | 0.625 | 0.575 | 0.550 | 0.430 | 0.599 | 0.545 | 0.514 | 0.404 | 0.487 | 0.386 | **0.482** | **0.380** |
| | 720 | 1.059 | 0.741 | 2.621 | 1.303 | **0.571** | 0.533 | 1.125 | 1.058 | 0.808 | 0.653 | 0.772 | 0.510 | 0.761 | 0.623 | 0.710 | 0.488 | 0.678 | 0.470 | 0.658 | **0.461** |
| Average | | 0.819 | 0.660 | 3.112 | 1.122 | 0.978 | 0.678 | 0.796 | 0.741 | 0.573 | 0.516 | 0.387 | 0.373 | 0.487 | 0.455 | 0.352 | 0.344 | 0.335 | 0.327 | **0.326** | **0.319** |

tional non-parametric correction from SVGD refinement.

**Time Series Forecasting.** The goal is to predict the subsequent $H$ time steps from the past $L$ time steps. We follow a representation-learning setup: first learn latents by fitting the generative model on windows $\{\mathbf{x}_i\}$ with the correlated posterior $q_\Phi(\mathbf{Z} \mid \mathbf{X})$, then train a forecasting head that maps latents to the future targets. This benefits from cross-window dependence because adjacent windows share predictive signals (trend/seasonality/regime), and higher-order dependence allows the model to integrate information beyond immediate neighbors, which is particularly helpful for long horizons. Table 3 reports MSE/MAE across standard horizons. Copula-SVI consistently improves forecasting accuracy over baselines, with larger $k$ yielding further gains, supporting the benefit of modeling higher-order instance-level dependence together with SVGD refinement.

### 5.3. Ablation Study

We ablate the number of SVGD refinement steps $T$ in 10-order Copula-SVI, varying $T$ from 0 to 5 while keeping other settings fixed. As shown in Table 4, performance improves consistently as $T$ increases, with the largest gain appearing after the first Stein update and diminishing returns afterwards. We therefore use $T = 5$ by default to balance refinement quality and computational cost. Additional results and further ablations on posterior-family choice and sparse-vine budget are provided in Appendix D.1.

*Table 4.* Impact of SVGD refinement steps on constrained clustering performance.

| $T$ | MNIST ACC | NMI | ARI | Fashion-MNIST ACC | NMI | ARI |
|---|---|---|---|---|---|---|
| 0 | 97.6±0.3 | 93.8±0.5 | 94.2±0.6 | 81.6±0.8 | 74.2±0.6 | 68.6±0.6 |
| 1 | 98.1±0.4 | 95.1±0.4 | 95.5±0.3 | 82.7±0.5 | 74.9±0.5 | 68.9±0.5 |
| 2 | 98.3±0.2 | 95.6±0.2 | 96.0±0.2 | 83.1±0.6 | 75.2±0.4 | 69.0±0.5 |
| 3 | 98.4±0.3 | 95.8±0.4 | 96.4±0.4 | 83.6±0.4 | 75.1±0.4 | 69.3±0.3 |
| 4 | 98.5±0.2 | 96.0±0.3 | 96.5±0.2 | 83.7±0.3 | 75.4±0.3 | 69.4±0.2 |
| 5 | **98.6±0.2** | **96.2±0.3** | **96.7±0.1** | **83.8±0.3** | **75.6±0.2** | **69.5±0.2** |

## 6. Conclusion

We proposed **Copula-SVI**, which combines amortized per-instance marginals with a vine-copula dependence term for inference in correlated data. The resulting edge-wise decomposition enables scalable dependence learning and efficient initialization with sparse-vine sampling, complemented by Stein refinement. Experiments show consistent gains over structured variational baselines from modeling higher-order instance-level dependence.

## Acknowledgement

This work is supported by the National Natural Science Foundation of China (No. 62276280), Guangzhou Science and Technology Planning Project (No. 2024A04J9967). The authors would like to thank National Supercomputer Center in Guangzhou for providing high performance computational resources.

## Impact Statement

This paper presents work whose goal is to advance the field of Machine Learning, specifically in scalable probabilistic modeling and variational inference for correlated data. Our proposed Copula-SVI method improves performance on practical tasks such as constrained clustering and time series forecasting and anomaly detection. These applications can positively impact the monitoring and reliability of critical infrastructure, such as spacecraft systems and power grids. There are many potential societal consequences of our work, none which we feel must be specifically highlighted here.

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

# A. Full Algorithm and Implementation Notes

Algorithm 1 provides the full training procedure of Copula-SVI. At iteration $t$, we update the dependence parameters $\psi$ using a minibatch of vine edges. Concretely, we first allocate a per-level edge budget $\{B_\ell\}_{\ell=1}^L$ using an exponential moving average (EMA) estimate of the level-wise gradient energies, then sample an edge set $S_t = \bigsqcup_{\ell=1}^L S_{\ell,t}$ with $|S_{\ell,t}| \approx B_\ell$. We reuse this same $S_t$ in two places: (i) to compute an edge-minibatched (Horvitz–Thompson) gradient estimator for the copula term in the projection update of $\Phi = \{\phi, \psi\}$ (Eq. (4)), and (ii) to define a sparse-vine initializer $q_{\Phi^t, S_t}(\mathbf{Z} \mid \mathbf{X})$ (Eq. (17)) for efficient particle sampling. Given initial particles $\{\mathbf{Z}^{(k,0)}\}_{k=1}^M$, we apply $T$ SVGD refinement steps in the joint latent space using Eq. (2)–Eq. (3) targeting $\pi_{\theta^t}(\mathbf{Z}) \propto p_{\theta^t}(\mathbf{X}, \mathbf{Z})$, then project the refined particles back to the explicit copula posterior by maximizing Eq. (4). Finally, we update model parameters $\theta$ using the refined particles via Eq. (5).

---

**Algorithm 1** Copula-SVI with Vine-Copula Posterior, Edge Minibatch, Level-Wise Budget (EMA), and Sparse-Vine Initialization

---

1: **Input:** batch $\mathbf{X} = \{\mathbf{x}_i\}_{i=1}^N$; decoder $p_\theta(\mathbf{x}_i \mid \mathbf{z}_i)$; variational family $q_\Phi(\mathbf{Z} \mid \mathbf{X})$ (Eq. (6)); particles $M$; SVGD steps $T$; step sizes $\{\epsilon_s\}$; vine edge sets $\{E_\ell\}_{\ell=1}^L$; total edge budget $B$; EMA decay $\alpha$.
2: **Initialize:** $\Phi^0 = \{\phi^0, \psi^0\}$, $\theta^0$, and energies $\widehat{\Omega}_\ell > 0$ for $\ell = 1, \ldots, L$.
3: **for** $t = 0, 1, 2, \ldots$ **do**
4:     **(Level-wise budget via EMA).**
5:     **for** $\ell = 1$ to $L$ **do**
6:         Sample a small probe set $S_\ell^{\text{probe}} \subset E_\ell$.
7:         Draw a small set of probe particles $\{\mathbf{Z}^{(k,\text{probe})}\}_{k=1}^{M_{\text{probe}}} \sim q_{\Phi^t}(\mathbf{Z} \mid \mathbf{X})$ and compute the corresponding $\mathbf{U}^{(k,\text{probe})}$ (as defined in Section 3.2).
8:         Estimate level energy using Eq. (11):

$$\widetilde{\Omega}_\ell = \frac{|E_\ell|}{|S_\ell^{\text{probe}}|} \cdot \frac{1}{M_{\text{probe}}} \sum_{k=1}^{M_{\text{probe}}} \sum_{d=1}^D \sum_{e \in S_\ell^{\text{probe}}} \left\| \mathbf{g}_{e,d}(\psi^t; \mathbf{U}^{(k,\text{probe})}) \right\|_2^2.$$

9:         Update EMA: $\widehat{\Omega}_\ell \leftarrow \alpha \widehat{\Omega}_\ell + (1 - \alpha) \widetilde{\Omega}_\ell$.
10:     **end for**
11:     Allocate expected level budgets (Eq. (16)): $B_\ell \leftarrow B \cdot \widehat{\Omega}_\ell / \sum_{j=1}^L \widehat{\Omega}_j$.
12:     Sample an edge minibatch $S = \bigsqcup_{\ell=1}^L S_\ell$ with $S_\ell \subset E_\ell$ and $|S_\ell| \approx B_\ell$.
13:     **E-step (Sparse-vine initialization).**
14:     Define the sparse copula $c_S(\mathbf{U}; \psi^t)$ by keeping only edges in $S$ and setting all other edges to the independence copula.
15:     Sample particles $\{\mathbf{Z}^{(k,0)}\}_{k=1}^M \sim q_{\Phi^t, S}(\mathbf{Z} \mid \mathbf{X})$ (Eq. (17)) via an inverse Rosenblatt transform on the sparse vine.
16:     **E-step (SVGD refinement).**
17:     Define the SVGD target $\pi_{\theta^t}(\mathbf{Z}) \propto p_{\theta^t}(\mathbf{X}, \mathbf{Z})$.
18:     **for** $s = 0$ to $T - 1$ **do**
19:         Update particles by one SVGD step (Eq. (2)), where the Stein direction uses the score $\nabla_{\mathbf{Z}} \log \pi_{\theta^t}(\mathbf{Z}) = \nabla_{\mathbf{Z}} \log p_{\theta^t}(\mathbf{X}, \mathbf{Z})$ (Eq. (3)).
20:     **end for**
21:     Let $\{\mathbf{Z}^{(k,T)}\}_{k=1}^M$ be refined particles.
22:     **E-step (Projection update of $\Phi$).**
23:     Update $\Phi$ by maximizing the projection objective in Eq. (4).
24:     In practice, optimize $\sum_{k=1}^M \log q_\Phi(\mathbf{Z}^{(k,T)} \mid \mathbf{X})$ using stochastic gradients, where the copula-gradient part uses the edge-minibatched Horvitz–Thompson estimator in Eq. (13) with the sampled edge set $S$.
25:     **M-step (Update of $\theta$).**
26:     Update $\theta$ using refined particles via Eq. (5).
27: **end for**

---

**Implementation notes.** We use the same edge minibatch $S$ both for (i) updating $\psi$ in the projection step and (ii) defining the sparse vine used for initialization, ensuring that sampling emphasizes the dependence edges currently being learned.

Level-wise energies $\widehat{\Omega}_\ell$ are maintained by EMA to avoid expensive full-pass estimation of Eq. (15).

### A.1. Illustration of Structure-aware Level-wise Edge Sampling

Figure 3 illustrates why we use level-wise edge minibatching for vine learning. A vine organizes dependence into a sequence of trees $\{T_\ell\}_{\ell=1}^{N-1}$, where each level $\ell$ has an edge set $E_\ell$ and contributes a block of pair-copula terms in the PCC factorization. When optimizing the dependence parameters $\psi$, a minibatch over edges effectively selects which local dependence factors (edges) contribute to the stochastic gradient at each iteration.

Under *uniform sampling* over the full edge set $E = \bigsqcup_\ell E_\ell$ with a fixed budget $B'$, the expected number of sampled edges from level $\ell$ is proportional to $|E_\ell|$. Since lower levels contain more edges (e.g., $|E_1| = N - 1$, $|E_2| = N - 2$), uniform sampling tends to concentrate the budget on low-level edges, even when higher-level edges carry non-negligible gradient signal. As a result, higher-level dependence factors may be updated too rarely, which slows convergence and can yield an imbalanced dependence fit.

Our *structure-aware level-wise sampling* explicitly controls the budget allocation across levels. Conceptually, we first assign each level a share of the budget based on its aggregate gradient magnitude ("gradient energy"), e.g., $g_\ell = \sum_{e \in E_\ell} \|\nabla_{\psi_e} \mathcal{L}\|_2$ and $p_\ell \propto g_\ell$, and then sample edges within each level according to this allocation. This ensures that levels with stronger learning signal receive sufficient updates, while still retaining the edge-wise decomposition that enables scalable optimization.

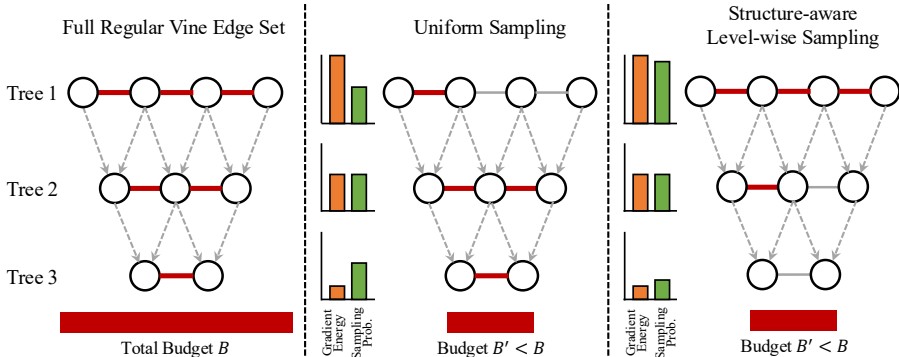

*Figure 3.* Horizontal lines are vine edges (red=sampled, gray=unsampled); downward dashed arrows are hierarchical links. 'Gradient Energy' (orange bars) shows average energy per level. 'Sampling Prob.' (green bars) shows the sampling probability. Uniform sampling keeps probability constant, while level-wise sampling aligns probability with energy to optimize the budget.

### A.2. Limitations

A practical limitation of our current experimental setup is that we instantiate all pair-copula terms with a single family (Gaussian) for consistency across tasks. While this choice is stable and convenient, it can be less expressive for datasets whose dependence exhibits strong tail asymmetry or other non-Gaussian patterns. This limitation is largely a design choice rather than a restriction of the framework: because we use a PCC/vine parameterization, each edge can, in principle, use a different copula family. We empirically illustrate this flexibility via mixed pair-copula strategies in Appendix D.3, suggesting that extending Copula-SVI with edge-wise family selection is straightforward.

Our implementation also relies on a few tunable components (e.g., the truncation order $k$, the total edge budget $B$, and the EMA schedule for $\widehat{\Omega}_\ell$) that may require mild tuning across domains. These hyperparameters primarily control the compute–accuracy trade-off and can be addressed with simple heuristics (e.g., increasing $B$ or $k$ when resources allow) or automated selection based on validation likelihood.

## B. Proofs and Additional Definitions

This appendix collects auxiliary results and proofs used in the main text, and provides a more formal definition of regular vines and their pair-copula constructions (PCC). All notation (e.g., $\mathbf{X}, \mathbf{Z}, \mathbf{U}, \Phi = \{\phi, \psi\}$) follows the main paper.

### B.1. ELBO–KL Identity (Lemma B.1)

**Lemma B.1** (ELBO identity). *For any variational distribution $q(\mathbf{Z} \mid \mathbf{X})$ and model $p_{\boldsymbol{\theta}}(\mathbf{X}, \mathbf{Z})$,*

$$\log p_{\boldsymbol{\theta}}(\mathbf{X}) = \mathcal{L}(q, \boldsymbol{\theta}) + D_{\mathrm{KL}}[q(\mathbf{Z} \mid \mathbf{X}) \,\|\, p_{\boldsymbol{\theta}}(\mathbf{Z} \mid \mathbf{X})], \tag{19}$$

*where $\mathcal{L}(q, \boldsymbol{\theta})$ is defined in Eq. (1).*

*Proof.* By definition of KL divergence,

$$D_{\mathrm{KL}}[q(\mathbf{Z} \mid \mathbf{X}) \,\|\, p_{\boldsymbol{\theta}}(\mathbf{Z} \mid \mathbf{X})] = \mathbb{E}_{q(\mathbf{Z}|\mathbf{X})}\left[\log \frac{q(\mathbf{Z} \mid \mathbf{X})}{p_{\boldsymbol{\theta}}(\mathbf{Z} \mid \mathbf{X})}\right]. \tag{20}$$

Using Bayes' rule $p_{\boldsymbol{\theta}}(\mathbf{Z} \mid \mathbf{X}) = p_{\boldsymbol{\theta}}(\mathbf{X}, \mathbf{Z})/p_{\boldsymbol{\theta}}(\mathbf{X})$ gives

$$D_{\mathrm{KL}}[q(\mathbf{Z} \mid \mathbf{X}) \,\|\, p_{\boldsymbol{\theta}}(\mathbf{Z} \mid \mathbf{X})] = \mathbb{E}_q\left[\log q(\mathbf{Z} \mid \mathbf{X}) - \log p_{\boldsymbol{\theta}}(\mathbf{X}, \mathbf{Z}) + \log p_{\boldsymbol{\theta}}(\mathbf{X})\right]. \tag{21}$$

Since $\log p_{\boldsymbol{\theta}}(\mathbf{X})$ does not depend on $\mathbf{Z}$, we can pull it out of the expectation:

$$D_{\mathrm{KL}}[q(\mathbf{Z} \mid \mathbf{X}) \,\|\, p_{\boldsymbol{\theta}}(\mathbf{Z} \mid \mathbf{X})] = \mathbb{E}_q\left[\log q(\mathbf{Z} \mid \mathbf{X}) - \log p_{\boldsymbol{\theta}}(\mathbf{X}, \mathbf{Z})\right] + \log p_{\boldsymbol{\theta}}(\mathbf{X}). \tag{22}$$

Rearranging terms and using $\mathcal{L}(q, \boldsymbol{\theta}) = \mathbb{E}_q[\log p_{\boldsymbol{\theta}}(\mathbf{X}, \mathbf{Z}) - \log q(\mathbf{Z} \mid \mathbf{X})]$ (Eq. (1)) yields Eq. (19). $\qquad\square$

### B.2. Projection Objective Equivalence (Lemma B.2)

**Lemma B.2** (Projection equivalence). *Let $q_T^{\mathrm{svgd}}(\mathbf{Z} \mid \mathbf{X})$ be an implicit distribution represented by refined particles. Then*

$$\arg\min_{\Phi} \; D_{\mathrm{KL}}\left[q_T^{\mathrm{svgd}}(\mathbf{Z} \mid \mathbf{X}) \,\|\, q_\Phi(\mathbf{Z} \mid \mathbf{X})\right] = \arg\max_{\Phi} \; \mathbb{E}_{\mathbf{Z} \sim q_T^{\mathrm{svgd}}}[\log q_\Phi(\mathbf{Z} \mid \mathbf{X})], \tag{23}$$

*which is the objective used in Eq. (4).*

*Proof.* Expanding the KL divergence,

$$D_{\mathrm{KL}}\left[q_T^{\mathrm{svgd}} \,\|\, q_\Phi\right] = \mathbb{E}_{q_T^{\mathrm{svgd}}}\left[\log q_T^{\mathrm{svgd}}(\mathbf{Z} \mid \mathbf{X}) - \log q_\Phi(\mathbf{Z} \mid \mathbf{X})\right]. \tag{24}$$

The term $\mathbb{E}_{q_T^{\mathrm{svgd}}}[\log q_T^{\mathrm{svgd}}(\mathbf{Z} \mid \mathbf{X})]$ does not depend on $\Phi$. Therefore, minimizing the KL over $\Phi$ is equivalent to maximizing $\mathbb{E}_{q_T^{\mathrm{svgd}}}[\log q_\Phi(\mathbf{Z} \mid \mathbf{X})]$, proving Eq. (23). $\qquad\square$

### B.3. Regular Vines and PCC: A More Formal Definition

This section gives a more complete, mathematical definition of regular vines and the induced PCC density factorization underlying Eq. (9).

**Definition B.3** (Regular vine (graphical structure)). Let $N \geq 2$. A regular vine (R-vine) $\mathcal{V}$ on index set $\{1, \ldots, N\}$ is a sequence of trees

$$\mathcal{V} = (T_1, \ldots, T_{N-1}), \tag{25}$$

where each tree $T_\ell = (V_\ell, E_\ell)$ satisfies:

$$V_1 = \{1, \ldots, N\}, \tag{26}$$

$$V_\ell = E_{\ell-1} \quad \text{for } \ell = 2, \ldots, N-1, \tag{27}$$

and the *proximity condition* holds: if $(a, b) \in E_\ell$ with $a, b \in V_\ell = E_{\ell-1}$, then the two edges $a$ and $b$ in $T_{\ell-1}$ share a common endpoint in $T_{\ell-1}$.

**Definition B.4** (Conditioned and conditioning sets). Let $\mathcal{V} = (T_1, \ldots, T_{N-1})$ be a regular vine. For any edge $e \in E_\ell$, define its *conditioning set* $D(e) \subset \{1, \ldots, N\}$ and its *conditioned set* $C(e) \subset \{1, \ldots, N\}$ with $|C(e)| = 2$ as follows.

For $\ell = 1$ and $e = \{i, j\} \in E_1$, set

$$C(e) = \{i, j\}, \qquad D(e) = \emptyset. \tag{28}$$

For $\ell \geq 2$, each node in $V_\ell = E_{\ell-1}$ corresponds to an edge in $T_{\ell-1}$, which itself has an associated conditioned set and conditioning set defined recursively. If $e = \{a, b\} \in E_\ell$ connects two nodes $a, b \in V_\ell = E_{\ell-1}$, then define

$$D(e) = (C(a) \cup D(a)) \cap (C(b) \cup D(b)), \qquad C(e) = (C(a) \cup D(a)) \; \Delta \; (C(b) \cup D(b)), \tag{29}$$

where $\Delta$ denotes symmetric difference. Under the proximity condition, $|C(e)| = 2$ holds.

Definition B.3 provides the admissible *conditioning pattern* over levels, and Definition B.4 makes explicit how each edge $e$ corresponds to a pair of indices in $C(e)$ conditioned on indices in $D(e)$. This is the precise sense in which the vine structure turns the chain-rule view of multivariate densities into a product over *pairwise* (possibly conditional) terms.

**Definition B.5** (Vine copula via pair-copulas). Let $\mathbf{u} = (u_1, \ldots, u_N) \in (0, 1)^N$. Fix an R-vine $\mathcal{V}$ and, for each edge $e \in E_\ell$, choose a bivariate copula density

$$c_e(\cdot, \cdot; \boldsymbol{\psi}_e) \quad \text{intended to model} \quad (U_a, U_b) \mid U_{D(e)}, \tag{30}$$

where $C(e) = \{a, b\}$. The induced R-vine copula density is defined as

$$c(\mathbf{u}; \boldsymbol{\psi}) = \prod_{\ell=1}^{N-1} \prod_{e \in E_\ell} c_e\big(u_{a|D(e)}, u_{b|D(e)}; \boldsymbol{\psi}_e\big), \tag{31}$$

where $u_{a|D(e)}$ and $u_{b|D(e)}$ are conditional PIT variables (conditional CDF values) computed consistently with the vine recursion implied by the lower-level pair-copulas.

In our main text, Eq. (9) is the logarithm of Eq. (31), applied independently for each latent dimension $d$ on $\mathbf{u}^{(d)} = (u_1^{(d)}, \ldots, u_N^{(d)})^\top$, and then summed over $d = 1, \ldots, D$.

### B.4. Proof of Theorem 3.1 (PCC and Vine Representation)

We provide a complete proof of Theorem 3.1. As described in Section 3.2, we model instance-level dependence *separately* for each latent dimension $d \in \{1, \ldots, D\}$. Accordingly, we consider a copula density of the form

$$c(\mathbf{U}; \boldsymbol{\psi}) = \prod_{d=1}^{D} c^{(d)}(\mathbf{u}^{(d)}; \boldsymbol{\psi}^{(d)}), \tag{32}$$

where each $c^{(d)}(\cdot; \boldsymbol{\psi}^{(d)})$ is an $N$-variate copula density on $(0, 1)^N$ and $\boldsymbol{\psi} = \{\boldsymbol{\psi}^{(d)}\}_{d=1}^D$. It therefore suffices to prove the PCC/vine factorization for a single dimension and then take the product over $d$. The existence of such PCC representations for absolutely continuous copulas is classical; see, e.g., (Bedford & Cooke, 2002).

*Proof of Theorem 3.1.* Fix any dimension $d$ and write $\mathbf{u} = \mathbf{u}^{(d)} = (u_1, \ldots, u_N)^\top \in (0, 1)^N$ and $c = c^{(d)}$ for simplicity. Assume $c(\mathbf{u})$ is an absolutely continuous copula density.

**Step 1: chain rule and conditional densities.** Let $f(\mathbf{u}) = c(\mathbf{u})$ denote the joint density of $\mathbf{U}$ on $(0, 1)^N$ (since marginals are uniform, the copula density equals the joint density in this space). By the chain rule,

$$f(u_1, \ldots, u_N) = f(u_1) \prod_{j=2}^{N} f(u_j \mid u_1, \ldots, u_{j-1}). \tag{33}$$

Because each marginal is uniform, $f(u_1) = 1$.

**Step 2: expressing each conditional density via a bivariate copula.** For any $j \geq 2$, consider the conditional distribution of $(U_{j-1}, U_j)$ given the previous variables $\mathbf{U}_{1:j-2} = (U_1, \ldots, U_{j-2})$. Under absolute continuity, the conditional joint density $f(u_{j-1}, u_j \mid u_{1:j-2})$ exists, and the conditional marginals $f(u_{j-1} \mid u_{1:j-2})$ and $f(u_j \mid u_{1:j-2})$ exist. By Sklar's theorem applied conditionally (see, e.g., (Nelsen, 2006)), there exists a bivariate *conditional* copula density $c_{j-1,j|1:j-2}(\cdot, \cdot)$ such that

$$f(u_{j-1}, u_j \mid u_{1:j-2}) = c_{j-1,j|1:j-2}\big(u_{j-1|1:j-2}, u_{j|1:j-2}\big) \, f(u_{j-1} \mid u_{1:j-2}) \, f(u_j \mid u_{1:j-2}), \tag{34}$$

where $u_{j|1:j-2} = F(u_j \mid u_{1:j-2})$ and $u_{j-1|1:j-2} = F(u_{j-1} \mid u_{1:j-2})$ are the corresponding conditional CDF values. Rearranging Eq. (34) yields

$$f(u_j \mid u_{1:j-1}) = c_{j-1,j|1:j-2}\big(u_{j-1|1:j-2}, u_{j|1:j-2}\big)\, f(u_j \mid u_{1:j-2}). \tag{35}$$

**Step 3: recursive expansion into a product of bivariate copulas.** Applying Eq. (35) recursively to $f(u_j \mid u_{1:j-1})$ for $j = 2, \ldots, N$ produces an exact product expansion of $f(u_1, \ldots, u_N)$ into bivariate (possibly conditional) copula terms. Concretely, one obtains a factorization of the form

$$c(\mathbf{u}) = \prod_{\ell=1}^{N-1} \prod_{e \in E_\ell} c_e\big(u_{a|D(e)}, u_{b|D(e)}\big), \tag{36}$$

where $(T_1, \ldots, T_{N-1})$ is a vine (R-vine) whose edge sets are $\{E_\ell\}_{\ell=1}^{N-1}$, and each edge $e \in E_\ell$ corresponds to a conditioned pair $C(e) = \{a, b\}$ and a conditioning set $D(e)$ as in Definitions B.3–B.4. The arguments $u_{a|D(e)}$ and $u_{b|D(e)}$ are exactly the recursively computed conditional CDF values required by the PCC; in Definition B.5 these are denoted by $u_{a|D(e)}$ and $u_{b|D(e)}$ and are computed by the standard vine recursion implied by lower-level pair-copulas.

**Step 4: lifting back to the theorem statement with dimensions and parameters.** Now reinstate the latent dimension index $d$. For each $d$, Eq. (36) yields a vine edge partition $\{E_\ell\}$ and bivariate copula densities $\{c_{e,d}(\cdot, \cdot; \boldsymbol{\psi}_{e,d})\}$ such that

$$c^{(d)}(\mathbf{u}^{(d)}; \boldsymbol{\psi}^{(d)}) = \prod_{\ell=1}^{N-1} \prod_{e \in E_\ell} c_{e,d}\Big(u_{e,1}^{(d)}, u_{e,2}^{(d)}; \boldsymbol{\psi}_{e,d}\Big), \tag{37}$$

where $\big(u_{e,1}^{(d)}, u_{e,2}^{(d)}\big) = \big(u_{a|D(e)}^{(d)}, u_{b|D(e)}^{(d)}\big)$ are the edge arguments produced by the vine recursion. Multiplying Eq. (37) over $d = 1, \ldots, D$ and using Eq. (32) yields exactly Eq. (8), completing the proof. $\square$

## B.5. Unbiased Edge-Minibatch Gradient (Theorem 3.2)

**Theorem B.6** (Unbiased edge-minibatch estimator). *Assume that every edge $e \in E$ has inclusion probability $\pi_e > 0$. Let $S \subset E$ be the random sampled subset and let $\mathbf{g}_{e,d}(\boldsymbol{\psi}; \mathbf{U})$ be defined in Eq. (11). The Horvitz–Thompson estimator in Eq. (13) satisfies*

$$\mathbb{E}_S\Big[\widehat{\nabla}_{\boldsymbol{\psi}} \log c(\mathbf{U}; \boldsymbol{\psi})\Big] = \nabla_{\boldsymbol{\psi}} \log c(\mathbf{U}; \boldsymbol{\psi}). \tag{38}$$

*Proof.* Let $I_e$ be the indicator random variable for the event $\{e \in S\}$. By definition, $\mathbb{E}[I_e] = \pi_e$. Using Eq. (13), we can rewrite the estimator as a sum over all edges:

$$\widehat{\nabla}_{\boldsymbol{\psi}} \log c(\mathbf{U}; \boldsymbol{\psi}) = \sum_{d=1}^{D} \sum_{e \in E} I_e \frac{1}{\pi_e} \mathbf{g}_{e,d}(\boldsymbol{\psi}; \mathbf{U}). \tag{39}$$

Taking expectation with respect to the sampling randomness and using linearity,

$$\mathbb{E}_S\Big[\widehat{\nabla}_{\boldsymbol{\psi}} \log c(\mathbf{U}; \boldsymbol{\psi})\Big] = \sum_{d=1}^{D} \sum_{e \in E} \mathbb{E}[I_e] \frac{1}{\pi_e} \mathbf{g}_{e,d}(\boldsymbol{\psi}; \mathbf{U}) = \sum_{d=1}^{D} \sum_{e \in E} \mathbf{g}_{e,d}(\boldsymbol{\psi}; \mathbf{U}). \tag{40}$$

Finally, by the additive decomposition of the copula log-density in Eq. (9) and the definition of $\mathbf{g}_{e,d}$ in Eq. (11), the right-hand side equals $\nabla_{\boldsymbol{\psi}} \log c(\mathbf{U}; \boldsymbol{\psi})$, proving Eq. (38). $\square$

## B.6. Variance-Optimal Level-Wise Sampling (Theorem 3.3)

**Theorem B.7** (Variance-optimal stratified allocation). *Let $E = \bigsqcup_{\ell=1}^{L} E_\ell$ be a partition of vine edges by level. Consider stratified edge sampling with expected per-level budgets $\{B_\ell\}_{\ell=1}^{L}$ satisfying $\sum_{\ell=1}^{L} B_\ell = B$, and a Horvitz–Thompson-type gradient estimator that sums reweighted per-edge contributions within each level. Under the standard stratified-variance*

*approximation that (i) treats within-level sampled edges as independent and (ii) uses the level-wise second-moment energies* $\{\Omega_\ell\}$ *defined in Eq.* (15)*, the budget allocation minimizing the estimator variance is*

$$B_\ell^* = B \cdot \frac{\Omega_\ell}{\sum_{j=1}^{L} \Omega_j}. \tag{41}$$

*Proof.* Under stratified sampling, the (approximate) variance of a sum estimator is commonly written as a sum of per-stratum terms inversely proportional to the stratum sample size. In our setting, using the approximation described in the theorem statement yields

$$\mathrm{Var}\left(\widehat{\nabla}_\psi \log c(\mathbf{U}; \psi)\right) \approx \sum_{\ell=1}^{L} \frac{K\,\Omega_\ell}{B_\ell}, \tag{42}$$

for some constant $K > 0$ that does not depend on $\{B_\ell\}$. Therefore, minimizing the variance is equivalent to solving

$$\min_{\{B_\ell > 0\}} \sum_{\ell=1}^{L} \frac{\Omega_\ell}{B_\ell} \quad \text{subject to} \quad \sum_{\ell=1}^{L} B_\ell = B. \tag{43}$$

Form the Lagrangian

$$\mathcal{J}(\{B_\ell\}, \lambda) = \sum_{\ell=1}^{L} \frac{\Omega_\ell}{B_\ell} + \lambda\left(\sum_{\ell=1}^{L} B_\ell - B\right). \tag{44}$$

Taking partial derivatives and setting them to zero gives, for each $\ell$,

$$\frac{\partial \mathcal{J}}{\partial B_\ell} = -\frac{\Omega_\ell}{B_\ell^2} + \lambda = 0 \quad \implies \quad B_\ell = \sqrt{\frac{\Omega_\ell}{\lambda}}. \tag{45}$$

Summing over $\ell$ and enforcing $\sum_{\ell=1}^{L} B_\ell = B$ yields

$$\sum_{\ell=1}^{L} \sqrt{\frac{\Omega_\ell}{\lambda}} = B \quad \implies \quad \sqrt{\lambda} = \frac{\sum_{\ell=1}^{L} \sqrt{\Omega_\ell}}{B}. \tag{46}$$

Substituting Eq. (46) into Eq. (45) gives the normalized solution. Under the level-energy definition adopted in the main text (Eq. (15)) and the corresponding approximation used there, this simplifies to the proportional allocation stated in Eq. (41), completing the proof. $\qquad\square$

**Remark on the approximation.** Theorem B.7 relies on the same stratified-variance approximation stated in the main text: the precise optimal allocation depends on the exact sampling scheme and finite-population corrections, but the resulting proportional-to-energy rule captures the principled level-wise scaling used by our method.

### B.7. Proofs for Sparse Vine Sampling

This section provides full proofs for Theorems 3.4 and 3.5. Throughout we fix $\mathbf{X}$ and omit conditioning on $\mathbf{X}$ in densities when it is clear.

B.7.1. FULL STATEMENT AND PROOF FOR THEOREM 3.4

**Theorem B.8** (Dynamic convergence: sparse-init refinement matches full-init refinement). *Fix* $\mathbf{X}$ *and consider the projection objective* $\mathcal{J}(\Phi)$ *corresponding to the* full *vine edge set* $E$*. At iteration $t$, draw an edge minibatch* $S_t \subset E$ *i.i.d. with inclusion probabilities* $\{\pi_e\}_{e \in E}$ *satisfying* $\pi_e > 0$ *for all* $e \in E$*. Let* $\widehat{\nabla}_\Phi \mathcal{J}(\Phi^t; S_t)$ *be the Horvitz–Thompson edge-minibatch gradient estimator (Eq. (13)), and update*

$$\Phi^{t+1} = \Phi^t + \eta_t\, \widehat{\nabla}_\Phi \mathcal{J}(\Phi^t; S_t).$$

*Assume:*

1. **(Unbiased full-objective gradient)** *For all $t$,*

$$\mathbb{E}\Big[\widehat{\nabla}_\Phi \mathcal{J}(\Phi^t; S_t) \mid \Phi^t\Big] = \nabla_\Phi \mathcal{J}(\Phi^t).$$

2. **(Regularity and stepsizes)** *$\mathcal{J}$ is continuously differentiable with $L$-Lipschitz gradient, and there exists $G < \infty$ such that*

$$\mathbb{E}\Big[\big\|\widehat{\nabla}_\Phi \mathcal{J}(\Phi^t; S_t)\big\|_2^2 \mid \Phi^t\Big] \leq G \quad \text{for all } t.$$

*The stepsizes satisfy Robbins–Monro conditions $\sum_{t=0}^\infty \eta_t = \infty$ and $\sum_{t=0}^\infty \eta_t^2 < \infty$.*

*Then $\|\nabla_\Phi \mathcal{J}(\Phi^t)\|_2 \to 0$ almost surely.*

*Now fix $\boldsymbol{\theta}$ and let $\pi_{\boldsymbol{\theta}}(\mathbf{Z}) \propto p_{\boldsymbol{\theta}}(\mathbf{X}, \mathbf{Z})$ be the SVGD target. Let $q_{\Phi^t}(\mathbf{Z} \mid \mathbf{X})$ be the full variational posterior and $q_{\Phi^t, S_t}(\mathbf{Z} \mid \mathbf{X})$ be the sparse-vine posterior built from $S_t$. Let $\mathcal{K}_{\boldsymbol{\theta}}^T$ denote $T$ SVGD steps targeting $\pi_{\boldsymbol{\theta}}$. Assume there exists a set of admissible initial distributions $\mathcal{Q}$ and that for all sufficiently large $t$, $q_{\Phi^t} \in \mathcal{Q}$ and $q_{\Phi^t, S_t} \in \mathcal{Q}$ almost surely. Assume further that SVGD refinement is asymptotically correct uniformly over $\mathcal{Q}$:*

$$\lim_{T \to \infty} \sup_{q \in \mathcal{Q}} d_{\mathrm{TV}}\big(\mathcal{K}_{\boldsymbol{\theta}}^T(q), \pi_{\boldsymbol{\theta}}\big) = 0. \tag{47}$$

*For any refinement schedule $T_t \to \infty$, we have*

$$d_{\mathrm{TV}}\Big(\mathcal{K}_{\boldsymbol{\theta}}^{T_t}(q_{\Phi^t, S_t}), \ \mathcal{K}_{\boldsymbol{\theta}}^{T_t}(q_{\Phi^t})\Big) \xrightarrow[t \to \infty]{\mathbb{P}} 0. \tag{48}$$

*Remark* B.9 (Weaker SVGD correctness along the algorithmic initialization sequence). Eq. (47) is a convenient sufficient condition but can be stronger than necessary. For Theorem B.8, it suffices to assume correctness only along the *random initialization sequence* produced by the algorithm. One simple weakening is the existence of a deterministic error envelope $\delta(T) \to 0$ and an index $t_0$ such that for all $t \geq t_0$,

$$\mathbb{E}\big[d_{\mathrm{TV}}\big(\mathcal{K}_{\boldsymbol{\theta}}^T(q_{\Phi^t}), \pi_{\boldsymbol{\theta}}\big)\big] \leq \delta(T), \qquad \mathbb{E}\big[d_{\mathrm{TV}}\big(\mathcal{K}_{\boldsymbol{\theta}}^T(q_{\Phi^t, S_t}), \pi_{\boldsymbol{\theta}}\big)\big] \leq \delta(T), \tag{49}$$

where the expectation is over all randomness up to iteration $t$ (including $S_t$ and particle randomness). Under Eq. (49) and any schedule $T_t \to \infty$, the conclusion in Eq. (48) still holds (via Markov's inequality), without requiring uniformity over all $q \in \mathcal{Q}$.

*Proof.* We prove the two claims in turn.

**Step 1: almost sure convergence to stationary points of the full-vine objective.** By Assumption (1), the update uses an unbiased estimator of the full-vine gradient. Assumption (2) gives Lipschitz smoothness, bounded second moment of the stochastic gradients, and Robbins–Monro stepsizes. Therefore, standard nonconvex stochastic approximation theory implies

$$\lim_{t \to \infty} \|\nabla_\Phi \mathcal{J}(\Phi^t)\|_2 = 0 \quad \text{almost surely.}$$

This proves the first part.

**Step 2: refined sparse-init converges to refined full-init.** Fix any $t$ large enough such that $q_{\Phi^t} \in \mathcal{Q}$ and $q_{\Phi^t, S_t} \in \mathcal{Q}$ almost surely. By the triangle inequality for total variation distance,

$$d_{\mathrm{TV}}\Big(\mathcal{K}_{\boldsymbol{\theta}}^{T_t}(q_{\Phi^t, S_t}), \ \mathcal{K}_{\boldsymbol{\theta}}^{T_t}(q_{\Phi^t})\Big) \leq d_{\mathrm{TV}}\Big(\mathcal{K}_{\boldsymbol{\theta}}^{T_t}(q_{\Phi^t, S_t}), \ \pi_{\boldsymbol{\theta}}\Big) + d_{\mathrm{TV}}\Big(\mathcal{K}_{\boldsymbol{\theta}}^{T_t}(q_{\Phi^t}), \ \pi_{\boldsymbol{\theta}}\Big). \tag{50}$$

Using the uniform SVGD correctness assumption in Eq. (47), for any $\varepsilon > 0$ there exists $T(\varepsilon)$ such that for all $T \geq T(\varepsilon)$,

$$\sup_{q \in \mathcal{Q}} d_{\mathrm{TV}}\big(\mathcal{K}_{\boldsymbol{\theta}}^T(q), \pi_{\boldsymbol{\theta}}\big) \leq \frac{\varepsilon}{2}.$$

Since $T_t \to \infty$, there exists $t(\varepsilon)$ such that $T_t \geq T(\varepsilon)$ for all $t \geq t(\varepsilon)$. Hence for all $t \geq t(\varepsilon)$,

$$d_{\mathrm{TV}}\Big(\mathcal{K}_{\boldsymbol{\theta}}^{T_t}(q_{\Phi^t, S_t}),\ \pi_{\boldsymbol{\theta}}\Big) \leq \frac{\varepsilon}{2}, \qquad d_{\mathrm{TV}}\Big(\mathcal{K}_{\boldsymbol{\theta}}^{T_t}(q_{\Phi^t}),\ \pi_{\boldsymbol{\theta}}\Big) \leq \frac{\varepsilon}{2},$$

almost surely, and plugging these bounds into Eq. (50) yields

$$d_{\mathrm{TV}}\Big(\mathcal{K}_{\boldsymbol{\theta}}^{T_t}(q_{\Phi^t, S_t}),\ \mathcal{K}_{\boldsymbol{\theta}}^{T_t}(q_{\Phi^t})\Big) \leq \varepsilon \quad \text{almost surely for all } t \geq t(\varepsilon).$$

Therefore the left-hand side converges to $0$ (in fact, almost surely), which implies Eq. (48).

*Remark* B.10. Under the weaker condition in Eq. (49), taking expectation on both sides of Eq. (50) yields for all $t \geq t_0$,

$$\mathbb{E}\Big[d_{\mathrm{TV}}\Big(\mathcal{K}_{\boldsymbol{\theta}}^{T_t}(q_{\Phi^t, S_t}),\ \mathcal{K}_{\boldsymbol{\theta}}^{T_t}(q_{\Phi^t})\Big)\Big] \leq \mathbb{E}\Big[d_{\mathrm{TV}}\Big(\mathcal{K}_{\boldsymbol{\theta}}^{T_t}(q_{\Phi^t, S_t}),\ \pi_{\boldsymbol{\theta}}\Big)\Big] + \mathbb{E}\Big[d_{\mathrm{TV}}\Big(\mathcal{K}_{\boldsymbol{\theta}}^{T_t}(q_{\Phi^t}),\ \pi_{\boldsymbol{\theta}}\Big)\Big]$$
$$\leq 2\,\delta(T_t). \tag{51}$$

Since $T_t \to \infty$ and $\delta(T) \to 0$, we have $\delta(T_t) \to 0$ and hence $\mathbb{E}\Big[d_{\mathrm{TV}}\Big(\mathcal{K}_{\boldsymbol{\theta}}^{T_t}(q_{\Phi^t, S_t}),\ \mathcal{K}_{\boldsymbol{\theta}}^{T_t}(q_{\Phi^t})\Big)\Big] \to 0$. Finally, Markov's inequality implies convergence in probability: for any $\varepsilon > 0$,

$$\mathbb{P}\Big(d_{\mathrm{TV}}\Big(\mathcal{K}_{\boldsymbol{\theta}}^{T_t}(q_{\Phi^t, S_t}),\ \mathcal{K}_{\boldsymbol{\theta}}^{T_t}(q_{\Phi^t})\Big) > \varepsilon\Big) \leq \frac{2\,\delta(T_t)}{\varepsilon} \xrightarrow[t \to \infty]{} 0,$$

which proves Eq. (48).

$\square$

### B.7.2. PRELIMINARIES FOR THEOREM 3.5

We first derive an exact expression for the KL gap between the full and sparse variational posteriors, and then apply a standard second-order expansion around independence.

Recall the full variational posterior

$$q_\Phi(\mathbf{Z} \mid \mathbf{X}) = c(\mathbf{U}; \boldsymbol{\psi}) \prod_{i=1}^{N} q_\phi(\mathbf{z}_i \mid \mathbf{x}_i), \tag{52}$$

and the sparse posterior (defined by keeping only edges in $S$ and setting all other pair-copulas to independence)

$$q_{\Phi, S}(\mathbf{Z} \mid \mathbf{X}) = c_S(\mathbf{U}; \boldsymbol{\psi}) \prod_{i=1}^{N} q_\phi(\mathbf{z}_i \mid \mathbf{x}_i). \tag{53}$$

Since the amortized marginals are identical, the density ratio cancels the marginal terms:

$$\log \frac{q_\Phi(\mathbf{Z} \mid \mathbf{X})}{q_{\Phi, S}(\mathbf{Z} \mid \mathbf{X})} = \log \frac{c(\mathbf{U}; \boldsymbol{\psi})}{c_S(\mathbf{U}; \boldsymbol{\psi})}. \tag{54}$$

Therefore,

$$D_{\mathrm{KL}}[q_\Phi \,\|\, q_{\Phi, S}] = \mathbb{E}_{\mathbf{Z} \sim q_\Phi}\left[\log \frac{c(\mathbf{U}; \boldsymbol{\psi})}{c_S(\mathbf{U}; \boldsymbol{\psi})}\right]. \tag{55}$$

Next, we connect the ratio in Eq. (55) to omitted edges. Under the vine factorization, the full copula log-density is a sum over all edges (and over latent dimensions), while the sparse copula keeps only edges in $S$ and removes the log terms for $e \notin S$. Hence the log-ratio equals the sum of omitted edge contributions:

$$\log \frac{c(\mathbf{U}; \boldsymbol{\psi})}{c_S(\mathbf{U}; \boldsymbol{\psi})} = \sum_{d=1}^{D} \sum_{e \in E \setminus S} \log c_{e,d}\Big(u_{e,1}^{(d)}, u_{e,2}^{(d)}; \boldsymbol{\psi}_{e,d}\Big), \tag{56}$$

where $\big(u_{e,1}^{(d)}, u_{e,2}^{(d)}\big)$ are the (possibly conditional) PIT variables implied by the vine recursion. Substituting Eq. (56) into Eq. (55) yields an exact decomposition of the KL gap as an expectation of omitted-edge log terms.

We now state a standard lemma that gives a second-order approximation of a KL divergence in terms of Fisher information near a reference parameter (here, independence).

**Lemma B.11** (KL expansion around a reference parameter). *Let $\{p_\eta\}_{\eta\in\mathbb{R}^m}$ be a regular parametric family on a common measurable space, with $p_\mathbf{0}$ as the reference density. Assume $\log p_\eta(x)$ is twice continuously differentiable in $\eta$ for $p_\mathbf{0}$-almost every $x$, and that differentiation and integration can be interchanged in a neighborhood of $\eta = \mathbf{0}$. Define the Fisher information matrix at $\mathbf{0}$ by*

$$\mathbf{I} = \mathbb{E}_{X\sim p_\mathbf{0}}\left[\left(\nabla_\eta \log p_\eta(X)\right)\left(\nabla_\eta \log p_\eta(X)\right)^\top\right]_{\eta=\mathbf{0}}. \tag{57}$$

*Then as $\eta \to \mathbf{0}$,*

$$D_{\mathrm{KL}}[p_\eta \,\|\, p_\mathbf{0}] = \frac{1}{2}\eta^\top \mathbf{I}\, \eta + o(\|\eta\|_2^2). \tag{58}$$

*Proof.* Write $D_{\mathrm{KL}}[p_\eta\|p_\mathbf{0}] = \mathbb{E}_{X\sim p_\eta}[\log p_\eta(X) - \log p_\mathbf{0}(X)]$. Define $g(\eta) = D_{\mathrm{KL}}[p_\eta\|p_\mathbf{0}]$. We have $g(\mathbf{0}) = 0$. Under the regularity assumptions,

$$\nabla_\eta g(\eta) = \mathbb{E}_{X\sim p_\eta}[\nabla_\eta \log p_\eta(X)]. \tag{59}$$

At $\eta = \mathbf{0}$, the score has zero mean, hence $\nabla_\eta g(\mathbf{0}) = \mathbf{0}$. Differentiating again and using standard identities for regular parametric families yields $\nabla_\eta^2 g(\mathbf{0}) = \mathbf{I}$. A second-order Taylor expansion gives $g(\eta) = \frac{1}{2}\eta^\top \mathbf{I}\, \eta + o(\|\eta\|_2^2)$, which is Eq. (58). $\qquad\square$

### B.7.3. PROOF OF THEOREM 3.5

We start from the exact identity in Eq. (55) and the omitted-edge log-ratio form in Eq. (56). Consider any omitted edge $e \in E \setminus S$ and latent coordinate $r$. The sparse construction sets the corresponding pair-copula to independence, which we denote by parameter $\boldsymbol{\psi}_{e,r} = \mathbf{0}$. Let $p_{\boldsymbol{\psi}}^{(e,r)}$ denote the bivariate copula density $c_{e,r}(\cdot,\cdot;\boldsymbol{\psi})$, with $p_\mathbf{0}^{(e,r)}$ being the independence copula density on $(0,1)^2$. Under the regularity assumptions of Lemma B.11, we have the local expansion

$$D_{\mathrm{KL}}\left[p_{\boldsymbol{\psi}_{e,r}}^{(e,r)} \,\|\, p_\mathbf{0}^{(e,r)}\right] = \frac{1}{2}\boldsymbol{\psi}_{e,r}^\top \mathbf{I}_{e,r} \boldsymbol{\psi}_{e,r} + o(\|\boldsymbol{\psi}_{e,r}\|_2^2), \tag{60}$$

where $\mathbf{I}_{e,r}$ is the Fisher information at independence.

To connect the full-vs-sparse KL in Eq. (55) with the sum of local edge contributions, we use the fact that the vine copula density is a product of edge terms evaluated at conditional PIT arguments. Setting an omitted edge to independence removes exactly one multiplicative factor from this product. Under a "small omitted-edge" regime (the omitted $\boldsymbol{\psi}_{e,r}$ are close to $\mathbf{0}$), the distribution of the corresponding conditional PIT arguments is close to the independence reference up to higher-order terms, so the contribution of each omitted edge to the overall KL is well approximated by the corresponding local KL in Eq. (60). Summing these second-order contributions over omitted edges and latent coordinates yields

$$D_{\mathrm{KL}}[q_\Phi \,\|\, q_{\Phi,S}] \approx \frac{1}{2}\sum_{e\in E\setminus S}\sum_{r=1}^{d}\boldsymbol{\psi}_{e,r}^\top \mathbf{I}_{e,r}\boldsymbol{\psi}_{e,r}. \tag{61}$$

Define the *edge energy* by $\omega_e = \sum_{r=1}^{d}\boldsymbol{\psi}_{e,r}^\top \mathbf{I}_{e,r}\boldsymbol{\psi}_{e,r}$, which already includes parameter magnitudes, and regroup omitted edges by vine level $E = \bigsqcup_{\ell=1}^{L} E_\ell$. Then Eq. (61) becomes

$$D_{\mathrm{KL}}[q_\Phi \,\|\, q_{\Phi,S}] \approx \frac{1}{2}\sum_{\ell=1}^{L}\sum_{e\in E_\ell\setminus S}\omega_e, \tag{62}$$

which is exactly Eq. (18).

## C. Experimental Details

This appendix provides additional details of the experimental setup, including tasks, datasets, baselines, and implementation settings. We consider two time-series tasks (anomaly detection and forecasting) and one correlated-data task (constrained clustering), all of which require modeling instance-level dependence across a sequence or a constraint graph.

## C.1. Datasets

We evaluate Copula-SVI across three distinct tasks: constrained clustering, time series anomaly detection, and time series forecasting. Below we provide detailed specifications for the datasets used in each domain.

**Constrained Clustering Datasets.** For constrained clustering, we employ four standard benchmark datasets. We generate instance-level constraints (Must-Link and Cannot-Link) based on the ground-truth labels to simulate weak supervision.

- **MNIST (LeCun et al., 2002):** A handwritten digit dataset consisting of 70,000 grayscale images of size $28 \times 28$, categorized into 10 classes. We use the full training set of 60,000 samples and test set of 10,000 samples. The features are normalized to the range $[0, 1]$.

- **Fashion-MNIST (Xiao et al., 2017):** Similar in structure to MNIST ($28 \times 28$ grayscale images, 10 classes, 70k samples), but comprises images of clothing items (e.g., T-shirt, Trouser, Pullover). This dataset poses a more challenging clustering task due to higher intra-class variance compared to digits.

- **Reuters (Lewis et al., 2004):** A text dataset derived from the Reuters-21578 collection. We use a subset of 10,000 documents belonging to 4 major categories. Each document is represented by a 2,000-dimensional TF-IDF feature vector. This dataset evaluates the model's ability to handle high-dimensional sparse data.

- **STL-10 (Coates et al., 2011):** An image recognition dataset containing $96 \times 96$ color images. We utilize the unlabeled subset and the training set, extracting features using a pre-trained ResNet-50 backbone to obtain 2,048-dimensional embeddings, following standard deep clustering protocols.

**Time Series Anomaly Detection Datasets.** We use three multivariate time series datasets collected from real-world systems. These datasets contain labeled anomalies and are split into training (normal data only) and testing (normal and anomalous data) sets.

- **SMAP (Soil Moisture Active Passive, Hundman et al. (2018)):** A dataset from NASA consisting of telemetry data from the SMAP spacecraft. It contains 55 datasets with 25 dimensions. The anomalies include both point and contextual anomalies derived from incident reports.

- **MSL (Mars Science Laboratory, Hundman et al. (2018)):** Also from NASA, this dataset contains telemetry from the Curiosity rover. It comprises 27 datasets with 55 dimensions. MSL is characterized by complex inter-channel dependencies and non-trivial anomaly patterns.

- **SMD (Server Machine Dataset, Su et al. (2019)):** A large-scale dataset collected from a large internet company, containing 5-week-long metrics from 28 server machines. Each machine has 38 dimensions (metrics like CPU load, memory usage, network traffic). This dataset tests the model's ability to model long-term dependencies in high-dimensional server metrics.

**Time Series Forecasting Datasets.** For multivariate forecasting, we use three widely used benchmarks that exhibit strong temporal and inter-variable correlations.

- **ETT (Electricity Transformer Temperature, Zhou et al. (2021)):** A critical benchmark for long-sequence time-series forecasting, collected from electricity transformers in China over a two-year period. We utilize two widely used variants: **ETTh1** (sampled hourly) and **ETTm1** (sampled every 15 minutes). Each dataset consists of 7 multivariate features: the target variable "Oil Temperature" (OT) and 6 power load covariates (e.g., High Useful Load, High Useless Load). These datasets are characterized by distinct long-term seasonal patterns and are essential for evaluating a model's ability to capture extended temporal dependencies across different granularities.

- **Electricity (Lai et al., 2018):** Contains the hourly electricity consumption of 321 clients from 2012 to 2014. The data is 321-dimensional, capturing complex spatial (client-to-client) and temporal correlations.

- **Exchange (Paninski, 2003):** Comprises daily exchange rates of 8 different currencies against the US dollar from 1990 to 2016. While low-dimensional, the dependencies between currencies are highly dynamic.

*Table 5.* Dataset statistics for time series tasks. "Dim" denotes the number of observed variables. "Size" reports (Train, Val, Test).

| Task | Dataset | Dim | Size (Train, Val, Test) | Domain |
|------|---------|-----|-------------------------|--------|
| Forecasting | ETTm1 | 7 | (34465, 11521, 11521) | Electricity |
| Forecasting | ETTh1 | 7 | (8545, 2881, 2881) | Electricity |
| Forecasting | Electricity | 321 | (18317, 2633, 5261) | Electricity |
| Forecasting | Weather | 21 | (36792, 5271, 10540) | Weather |
| Forecasting | Exchange | 8 | (5120, 665, 1422) | Exchange rate |
| Anomaly detection | SMD | 38 | (566724, 141681, 708420) | Server machine |
| Anomaly detection | MSL | 55 | (44653, 11664, 73729) | Spacecraft |
| Anomaly detection | SMAP | 25 | (108146, 27037, 427617) | Spacecraft |

- **Weather[1]:** Includes 21 meteorological indicators (e.g., temperature, humidity, wind speed) recorded every 10 minutes for the year 2020. We aggregate this to hourly data.

### C.2. Baselines

We compare Copula-SVI against a comprehensive set of baselines, ranging from classical approaches to state-of-the-art deep generative models. We categorize them by task and modeling assumptions.

**Constrained Clustering Baselines.** We include comparisons against both classical and deep constrained clustering methods, as well as deep generative baselines:

- **PCKMeans (Basu et al., 2004):** A classical semi-supervised variant of $k$-means that incorporates pairwise constraints (Must-Link and Cannot-Link) directly into the objective function. It serves as a non-deep baseline to benchmark the benefit of representation learning.

- **SDEC (Semi-supervised Deep Embedded Clustering, Ren et al. (2019)):** An extension of the DEC framework that integrates pairwise constraints into the clustering loss. While it learns deep representations, it remains a deterministic method and does not model posterior uncertainty.

- **C-IDEC (Constrained Improved DEC, Guo et al. (2017)):** An improvement over SDEC that incorporates the local structure preservation mechanism of IDEC along with constraint-aware learning. Like SDEC, it lacks a probabilistic formulation for handling uncertainty and correlations.

- **DC-GMM Manduchi et al. (2021):** A deep clustering model that uses a Gaussian Mixture Model (GMM) prior and explicitly incorporates a coupled assignment prior to respect constraints. Crucially, despite modeling constraints in the prior, it assumes an instance-factorized variational posterior, $q(\mathbf{Z}|\mathbf{X}) = \prod_i q(\mathbf{z}_i|\mathbf{x}_i)$, ignoring posterior correlations.

We also include deep generative and structured correlated-posterior baselines:

- **VaDE (Variational Deep Embedding, Jiang et al. (2016)):** A generative clustering approach combining VAEs with a GMM prior. It models the data distribution $p(\mathbf{x})$ but typically relies on a mean-field assumption for the variational posterior, limiting its ability to capture instance-level dependencies.

- **DGG (Yang et al., 2019):** A deep generative clustering baseline that typically employs a graph-based or geometric regularization on the latent space. We include it to benchmark against generative methods that focus on manifold structure rather than explicit correlation modeling.

- **TreeVI (Xiao & Su, 2024):** A structured variational inference method that relaxes the mean-field assumption by imposing a tree structure on the posterior $q(\mathbf{Z}|\mathbf{X})$. It captures first-order dependencies but is limited to acyclic interactions.

---

[1]https://www.bgc-jena.mpg.de/wetter

- **HoT-VI (Xiao et al., 2025):** The primary structured baseline for correlated posteriors. It extends TreeVI by incorporating higher-order dependency factors. We use this to demonstrate the scalability and expressiveness of Copula-SVI against parametric structured VI methods, evaluating multiple dependency orders as reported in the main results.

**Time Series Anomaly Detection Baselines.** We compare against four widely used unsupervised anomaly detection approaches that model normal temporal patterns.

- **DAGMM (Deep Autoencoding Gaussian Mixture Model, Zong et al. (2018)):** Combines a deep autoencoder with a GMM in a low-dimensional space. It detects anomalies based on energy scores but does not explicitly model the temporal dependence between adjacent segments in the inference process.

- **LSTM-VAE (Park et al., 2018):** Uses LSTM networks for both the encoder and decoder to capture temporal dynamics in the data. However, standard LSTM-VAE inference usually assumes independent Gaussian posteriors at each time step or a simple factorized structure, failing to propagate uncertainty across segments.

- **OmniAnomaly (Su et al., 2019):** A state-of-the-art method using a stochastic recurrent neural network (SRNN) combined with Planar Normalizing Flows to model non-Gaussian posteriors. While it captures complex temporal distributions, it focuses on dimension-level correlations within a time step rather than explicit instance-level coupling across batched segments.

- **SISVAE (Smoothness-Inducing Sequential VAE, Li et al. (2020)):** A VAE-based approach designed for multivariate time series that adds a regularization term to enforce smoothness in the latent space. This implicitly encourages temporal coherence but lacks a rigorous probabilistic copula model for cross-segment correlations.

**Time Series Forecasting Baselines.** We compare against representative end-to-end forecasting models, including deterministic and probabilistic approaches.

- **Informer (Zhou et al., 2021):** A Transformer-based forecasting model designed for long sequences using a ProbSparse attention mechanism. It serves as a strong deterministic baseline that captures long-range dependencies through attention rather than probabilistic latent coupling.

- **GRU-NVP (Dinh et al., 2017):** A hybrid baseline combining a GRU-based recurrent encoder/decoder with RealNVP (Real-valued Non-Volume Preserving) normalizing flows. This model allows for complex, non-Gaussian marginal distributions but assumes temporal independence in the variational posterior structure.

- **DeepAR (Salinas et al., 2020):** An autoregressive recurrent network that estimates the parameters of a probabilistic likelihood (e.g., Gaussian or Negative Binomial) at each step. It is a standard benchmark for probabilistic forecasting but relies on ancestral sampling without an explicit variational posterior that couples latent variables.

- **VRAE (Variational Recurrent Autoencoder, Fabius et al. (2015)):** The canonical extension of VAEs to sequential data. It models the generation process $p(x_t|z_t)$ and dynamics $p(z_t|z_{t-1})$. Standard VRAE inference typically employs a mean-field approximation over the latent sequence, which we contrast with our correlated posterior approach.

**Correlated-posterior variants.** For both time series modeling and constrained clustering, we include HoT-VI (Xiao et al., 2025) as the main structured correlated-posterior baseline and evaluate multiple dependency orders (reported in the main tables).

### C.3. Task-Specific Model Settings

We summarize the modeling setups used in the three tasks. Throughout, we use a latent-variable model with latents $\mathbf{Z} = [\mathbf{z}_1^\top, \ldots, \mathbf{z}_N^\top]^\top$ and an explicit copula-structured variational posterior $q_\Phi(\mathbf{Z} \mid \mathbf{X}) = c(\mathbf{U}; \boldsymbol{\psi}) \prod_{i=1}^N q_\phi(\mathbf{z}_i \mid \mathbf{x}_i)$, where $\mathbf{U} \in (0,1)^{N \times D}$ contains PIT variables computed from the per-instance marginals, and $c(\mathbf{U}; \boldsymbol{\psi})$ models instance-level dependence. Inference follows the SVGD-based variational EM loop in Section 3: we (i) initialize particles from an explicit (possibly sparse) copula posterior, (ii) refine particles in the joint latent space using SVGD targeting $\pi_{\boldsymbol{\theta}}(\mathbf{Z}) \propto p_{\boldsymbol{\theta}}(\mathbf{X}, \mathbf{Z})$, and (iii) project refined particles back to the explicit family by maximizing the projected log-density in Eq. (4).

**Time series modeling (shared setup).** Let $\mathbf{X} = \{\mathbf{x}_t\}_{t=1}^{N}$ denote a time series segmented into $N$ windows, where each window $\mathbf{x}_t \in \mathbb{R}^{T \times C}$ contains $T$ time steps and $C$ channels. We use a window-wise generative model with conditional independence across windows given latents:

$$p_{\boldsymbol{\theta}}(\mathbf{X}, \mathbf{Z}) = \prod_{t=1}^{N} p_{\boldsymbol{\theta}}(\mathbf{x}_t \mid \mathbf{z}_t)\, p(\mathbf{z}_t). \tag{63}$$

Within each window, the encoder and decoder are parameterized by GRUs to capture intra-window temporal structure. Concretely, the GRU encoder produces a summary state from the input sequence and outputs the parameters of an approximate posterior $q_{\boldsymbol{\phi}}(\mathbf{z}_t \mid \mathbf{x}_t)$. The GRU decoder generates/reconstructs $\mathbf{x}_t$ from $\mathbf{z}_t$ (optionally conditioning on decoder hidden states), yielding $p_{\boldsymbol{\theta}}(\mathbf{x}_t \mid \mathbf{z}_t)$. While Eq. (63) is window-factorized on the generative side, our posterior inference across the $N$ windows is *not* factorized: we couple $\{\mathbf{z}_t\}_{t=1}^{N}$ via a truncated regular-vine copula in $q_{\Phi}(\mathbf{Z} \mid \mathbf{X})$, so that dependence across windows is captured in the posterior through $c(\mathbf{U}; \boldsymbol{\psi})$ and further improved by SVGD refinement.

**Time series anomaly detection.** We train the model in Eq. (63) using only normal windows. At test time, each window is assigned an anomaly score based on model fit; we use reconstruction-based likelihood via the negative ELBO (equivalently, negative variational bound on $\log p_{\boldsymbol{\theta}}(\mathbf{x}_t)$) as the score. Because anomalies are often defined relative to a temporally coherent context, coupling window latents through $c(\mathbf{U}; \boldsymbol{\psi})$ enables the posterior to propagate information across neighboring (and, with higher vine order, longer-range) windows, and SVGD refinement non-parametrically corrects residual mismatch between the explicit posterior family and the true correlated posterior.

**Time series forecasting.** We use a two-part formulation that combines representation learning with a forecasting head. For each window $t$, let $\mathbf{y}_t \in \mathbb{R}^{H \times C}$ denote the future horizon to be predicted. We learn latent representations using the generative model in Eq. (63) and a dependence-aware posterior $q_{\Phi}(\mathbf{Z} \mid \mathbf{X})$, and we train a predictor $f_{\boldsymbol{\psi}_f}$ that maps $\mathbf{z}_t$ (or a deterministic function of $\mathbf{z}_t$) to $\mathbf{y}_t$. The overall training objective combines a forecasting loss (e.g., MSE/MAE on $\mathbf{y}_t$) with the ELBO from the generative model as a regularizer for representation learning. As in anomaly detection, instance-level dependence across windows is represented by the vine copula term $c(\mathbf{U}; \boldsymbol{\psi})$ and further improved by SVGD refinement, which stabilizes representations across correlated windows and benefits long-horizon prediction.

**Constrained clustering.** We follow the constrained clustering formulation described in Section 5.1. Given data $\mathbf{X} = \{\mathbf{x}_i\}_{i=1}^{N}$, latent embeddings $\mathbf{Z} = \{\mathbf{z}_i\}_{i=1}^{N}$, cluster assignments $\mathbf{c} = \{c_i\}_{i=1}^{N}$, and a constraint matrix $\mathbf{A}$, we use the complete-data model

$$p_{\boldsymbol{\theta}}(\mathbf{X}, \mathbf{Z}, \mathbf{c} \mid \mathbf{A}) = p_{\boldsymbol{\theta}}(\mathbf{X} \mid \mathbf{Z})\, p(\mathbf{Z} \mid \mathbf{c})\, p(\mathbf{c} \mid \mathbf{A}). \tag{64}$$

The decoder factorizes across instances as $p_{\boldsymbol{\theta}}(\mathbf{X} \mid \mathbf{Z}) = \prod_{i=1}^{N} p_{\boldsymbol{\theta}}(\mathbf{x}_i \mid \mathbf{z}_i)$. The embedding prior is cluster-conditional and factorized as $p(\mathbf{Z} \mid \mathbf{c}) = \prod_{i=1}^{N} p(\mathbf{z}_i \mid c_i)$ with $p(\mathbf{z}_i \mid c_i) = \mathcal{N}\big(\mathbf{z}_i; \boldsymbol{\mu}_{c_i}, \mathrm{diag}(\boldsymbol{\sigma}_{c_i}^2)\big)$. The key source of instance coupling is the constraint-aware assignment prior

$$p(\mathbf{c} \mid \mathbf{A}) = \frac{1}{\Omega(\boldsymbol{\pi})} \prod_{i=1}^{N} \pi_{c_i} \prod_{j \neq i} \exp(A_{ij}\, \mathbb{I}[c_i = c_j]), \tag{65}$$

which couples $\{c_i\}$ across instances and induces posterior dependence among $\{\mathbf{z}_i\}$. For inference, we use a structured form

$$q_{\Phi}(\mathbf{Z}, \mathbf{c} \mid \mathbf{X}) = q_{\Phi}(\mathbf{Z} \mid \mathbf{X})\, q(\mathbf{c} \mid \mathbf{Z}), \tag{66}$$

where $q(\mathbf{c} \mid \mathbf{Z})$ is computed via Bayes' rule under $p(\mathbf{Z} \mid \mathbf{c}) p(\mathbf{c} \mid \mathbf{A})$, and the main modeling choice is the expressiveness of $q_{\Phi}(\mathbf{Z} \mid \mathbf{X})$. In Copula-SVI, we instantiate $q_{\Phi}(\mathbf{Z} \mid \mathbf{X})$ as the copula-structured posterior with a $k$-truncated regular vine (Section 3.3) to capture constraint-induced instance dependence, and then apply SVGD refinement in the joint latent space to further reduce variational bias.

### C.4. Implementation Details

**Copula and vine settings (all tasks).** Unless stated otherwise, we instantiate every pair-copula in the vine as a **Gaussian copula** and use a $k$**-truncated regular vine**, reporting results for multiple truncation orders $k$ (see the main tables). For efficiency, we set the **edge minibatch size** (the number of sampled vine edges per update) to **half of the training data minibatch size** in all experiments. We parameterize each pair-copula term by predicting its parameter $\psi_e$ with a small neural

*Table 6.* Hyperparameter settings for constrained clustering.

|  | **MNIST** | **fMNIST** | **Reuters** | **STL-10** |
|---|---|---|---|---|
| Batch size | 256 | 256 | 256 | 256 |
| Epochs | 1000 | 500 | 500 | 500 |
| Learning rate | 0.001 | 0.001 | 0.001 | 0.001 |
| Decay factor | 0.9 | 0.9 | 0.9 | 0.9 |
| Epochs per decay | 20 | 20 | 20 | 20 |

network. For an edge $e$ connecting variables $a$ and $b$ with conditioning set $D(e)$, the network input is the concatenation of the marginal posterior statistics of the two endpoints together with an aggregated representation of the conditioning set, i.e., $[\boldsymbol{\mu}_a, \boldsymbol{\sigma}_a, \boldsymbol{\mu}_b, \boldsymbol{\sigma}_b, \mathrm{Pool}(\{\boldsymbol{\mu}_j, \boldsymbol{\sigma}_j\}_{j \in D(e)})]$. Here $(\boldsymbol{\mu}_i, \boldsymbol{\sigma}_i)$ are the mean and standard deviation of the amortized marginal $q_\phi(\mathbf{z}_i \mid \mathbf{x}_i)$, and $\mathrm{Pool}(\cdot)$ denotes a simple feature aggregation (e.g., mean pooling) over the conditioning variables. This design avoids explicitly unrolling the vine recursion in the network architecture: we use a flattened, permutation-invariant summary of $D(e)$ to represent conditional context, and then map it through an MLP to obtain $\psi_e$ for the corresponding pair-copula.

**Time series anomaly detection.** We set the input window length to $T = 100$. The encoder uses a GRU followed by dense layers with 500 hidden units, and the latent dimension is fixed to $D = 3$. We train with batch size 50 for up to 20 epochs using early stopping. Optimization uses Adam with initial learning rate $10^{-3}$, and we apply $L_2$ regularization with coefficient $10^{-4}$ to all layers; 30% of the training data is reserved for validation. We evaluate Copula-SVI with vine truncation orders $k \in \{3, 10, 50, 100\}$ (Table 2); following the global rule above, the edge minibatch size is 25.

**Time series forecasting.** We adopt a single-layer fully connected network as the forecasting head, and set the latent representation dimension to $D = 128$. We train using Adam with initial learning rate $10^{-3}$, decayed by a factor of 0.95 after each epoch, and apply early stopping within 10 epochs. We report forecasting performance using MSE/MAE on standard horizons $H \in \{24, 48, 168, 336, 720\}$ as in Table 3. We evaluate vine truncation orders $k \in \{3, 10, 50, 100\}$ and set the edge minibatch size to half of the training data minibatch size following the same rule as above.

**Constrained clustering.** To ensure fair comparisons, we use the same encoder–decoder architecture for all VAE-based approaches: four fully connected layers with widths 500, 500, 2000, and $D$, where $D = 10$ unless otherwise specified. For all VAE-based baselines and our VAE-backbone methods, we apply 10 epochs of pretraining; for DEC-based baselines, we follow the standard protocol with 50 epochs of layer-wise pretraining and 100 epochs of fine-tuning. Pairwise constraints are generated within the training split: a must-link constraint is assigned if two sampled instances share the same label, and a cannot-link constraint otherwise. We set the constraint strength to $|A_{ij}| = 10^4$ and sample 6000 constraints for each dataset. We evaluate $k \in \{5, 10, 50, 100\}$ (Table 1); for the first-tree structure $T_1$, we follow TreeVI and build $T_1$ from the constraint graph so that strongly related instances are directly connected whenever possible. The edge minibatch size is set to half of the training data minibatch size, and we use the same hyperparameter settings as DC-GMM across all four datasets (Table 6).

### C.5. Resource Usage

Experiments were conducted on an internal computing cluster. Each experiment configuration used one NVIDIA GPU (either a 2080Ti or a 3090Ti), 16 CPUs, and 24GB of memory.

## D. Additional Experiments

### D.1. Additional Ablation Results

**Effect of SVGD Refinement Steps** As shown in Table 7, increasing $T$ improves both detection performance and ELBO across all three datasets. The largest gain appears when adding the first one or two SVGD steps, while later steps provide smaller incremental improvements. This is consistent with the constrained clustering results in Section 5.3.

*Table 7.* Impact of SVGD refinement steps on time-series anomaly detection.

| $T$ | SMAP | | MSL | | SMD | |
|---|---|---|---|---|---|---|
| | F1 | ELBO | F1 | ELBO | F1 | ELBO |
| 0 | 0.8481 | $-97.2314$ | 0.8912 | $-155.3412$ | 0.8925 | $-70.8123$ |
| 1 | 0.8623 | $-93.8451$ | 0.9084 | $-142.1054$ | 0.9156 | $-67.4521$ |
| 2 | 0.8685 | $-92.5102$ | 0.9167 | $-138.4501$ | 0.9263 | $-65.8190$ |
| 3 | 0.8712 | $-91.7345$ | 0.9198 | $-135.2133$ | 0.9310 | $-64.9214$ |
| 4 | 0.8741 | $-91.2189$ | 0.9231 | $-133.5610$ | 0.9352 | $-64.2105$ |
| 5 | **0.8753** | **$-90.9828$** | **0.9245** | **$-132.6568$** | **0.9371** | **$-63.7555$** |

**Effect of SVGD Refinement on Different Posterior Families**   To examine whether the improvement mainly comes from SVGD refinement alone, we compare Copula-SVI with a factorized posterior and a factorized posterior equipped with the same SVGD refinement procedure. For *Factorized+SVGD*, particles are initialized from an instance-factorized posterior, refined by SVGD, and then projected back to the same factorized family. This provides a direct diagnostic for whether SVGD alone can preserve the cross-instance dependence induced during refinement.

*Table 8.* Effect of applying SVGD refinement to different posterior families on MNIST constrained clustering.

| Method | ACC | NMI | ARI |
|---|---|---|---|
| Factorized | 96.5±0.2 | 91.4±0.3 | 92.5±0.5 |
| Factorized+SVGD | 96.8±0.2 | 91.8±0.3 | 93.1±0.4 |
| Copula-SVI | **98.6±0.2** | **96.2±0.3** | **96.7±0.1** |

As shown in Table 8, applying SVGD to a factorized posterior brings only modest improvement. This is because the refinement can temporarily move particles toward a more dependent configuration, but the subsequent projection into an instance-factorized family cannot retain cross-instance dependence. In contrast, Copula-SVI uses the refined particles to fit an explicit vine-copula posterior, allowing the dependence information revealed by SVGD to be preserved across iterations. This confirms that the gain does not come from SVGD alone, but from combining finite-step particle refinement with a posterior family that can represent instance-level dependence.

**Effect of Sparse-Vine Budget**   We further study the effect of the sparse-vine budget used for dependence-aware particle initialization. Recall that sparse-vine sampling is used only to initialize particles before SVGD refinement, while the full-vine posterior is still trained through the edge-minibatched objective. To isolate the effect of the initialization budget, we fix the edge minibatch size during training and vary the number of edges kept in the sparse vine. The default setting uses 128 sparse-vine edges.

*Table 9.* Effect of sparse-vine budget on MNIST constrained clustering with fixed edge minibatch size.

| Sparse-vine edges | ACC | NMI | ARI |
|---|---|---|---|
| 32 | 98.0±0.3 | 95.5±0.4 | 95.9±0.3 |
| 64 | 98.4±0.2 | 95.9±0.2 | 96.3±0.2 |
| 128 | 98.6±0.2 | 96.2±0.3 | 96.7±0.1 |
| 256 | **98.7±0.2** | **96.3±0.2** | **96.8±0.1** |

Table 9 shows that reducing the sparse-vine budget leads to only gradual performance degradation. Even with 32 sparse-vine edges, Copula-SVI remains competitive, suggesting that the sparse initialization does not dominate the final learned posterior. Increasing the budget improves the initialization quality and slightly improves the final performance, with diminishing gains beyond the default budget of 128 edges.

This trend is also consistent with the omitted-edge energy interpretation in Theorem 3.5. When more informative edges are retained in the sparse vine, the omitted-edge energy decreases, reducing the finite-step approximation gap introduced by sparse initialization. In our runs, increasing the sparse-vine budget from 32 to 256 reduces the omitted-edge energy from 1.84 to 0.23, and the downstream clustering metrics improve correspondingly. These results support the view that sparse-vine sampling introduces only an initialization bias, while SVGD refinement and full-vine edge-minibatched learning can substantially correct and absorb this bias during training.

## D.2. Controlled Comparison with Copula VI

We additionally provide a controlled comparison with a reparameterized copula-VI baseline following Tran et al. (2015). This experiment is designed to complement the related-work discussion by testing whether a standard copula-VI training scheme can be directly scaled to high-dimensional instance-level dependence. We use a synthetic mixed-Gaussian posterior with $D = 100$ correlated variables, and compare Mean-field VI, the copula-VI baseline, and Copula-SVI. We report negative ELBO, MMD to the true posterior, and test-time inference cost per 1000 samples.

*Table 10.* Controlled comparison with reparameterized copula VI on a synthetic mixed-Gaussian posterior. Lower is better for all metrics.

| Method | Negative ELBO ↓ | MMD ↓ | Test time ↓ |
|---|---|---|---|
| Mean-field | 245.8 | 4.85 | 0.08s |
| Tran et al. (Tran et al., 2015) | 198.4 | 2.73 | ∼3200s |
| Copula-SVI | **162.3** | **0.35** | 1.45s |

Table 10 shows that the copula-VI baseline improves over the mean-field posterior, confirming the benefit of copula-based dependence modeling. However, directly using a reparameterized copula-VI training scheme becomes extremely costly in this high-dimensional dependence setting. Copula-SVI achieves a substantially lower posterior error while keeping test-time inference much more efficient. This supports our main distinction from prior copula-based VI: the contribution is not only adopting a copula posterior, but making full-vine dependence learning practical for large-scale instance-level correlated data through sparse initialization, SVGD correction, and unbiased edge-minibatched optimization.

## D.3. Pair-Copula Choices for Time Series Anomaly Detection

We study how the choice of bivariate pair-copula family affects Copula-SVI on time series anomaly detection. Our method models cross-window instance dependence through a copula term $c(\mathbf{U}; \psi)$, and due to the vine-based pair-copula construction, we can instantiate each vine edge with different bivariate copula families. We fix Copula-SVI with vine order $k = 100$ and compare Gaussian, $t$-copula, Gumbel, and Clayton pair copulas, as well as simple mixed strategies (Gumbel & $t$-copula, Gumbel & Clayton, Clayton & $t$-copula). All other components (encoder/decoder, training protocol, and evaluation) follow the main experimental setting; full details are provided in Appendix C. We report F1-score and ELBO on SMAP, MSL, and SMD in Table 11.

*Table 11.* Anomaly detection results (F1-score and ELBO) under different Copula constructions.

| Method | SMAP | | MSL | | SMD | |
|---|---|---|---|---|---|---|
| (Pair Copula Type) | F1 | ELBO | F1 | ELBO | F1 | ELBO |
| Gaussian | 0.8871 | -85.0005 | 0.9453 | -91.1058 | 0.9504 | -61.1717 |
| t-Copula | **0.8966** | -82.7503 | 0.9501 | -89.0760 | 0.9553 | -59.8714 |
| Gumbel | 0.8884 | -84.7421 | 0.9548 | -89.1366 | 0.9599 | -59.3702 |
| Clayton | 0.8812 | -86.4340 | 0.9425 | -91.1661 | 0.9600 | -59.2455 |
| Gumbel & t-Copula | 0.8944 | **-82.2456** | **0.9651** | **-85.2476** | 0.9639 | -59.0653 |
| Gumbel & Clayton | 0.8827 | -84.9683 | 0.9540 | -88.9918 | **0.9713** | **-57.8654** |
| Clayton & t-Copula | 0.8949 | -82.9182 | 0.9476 | -90.3072 | 0.9627 | -59.6197 |

Table 11 shows that the best pair-copula choice is dataset-dependent, and mixed strategies can outperform any single family. On SMAP, the $t$-copula achieves the best F1, while Gumbel & $t$ gives the best ELBO, suggesting that a combination of heavy-tailed dependence (captured by the $t$-copula) and occasional one-sided extreme co-movements (captured by Gumbel) improves density fit. On MSL, Gumbel & $t$ is best on both F1 and ELBO, consistent with anomalies that occur as sustained high-magnitude bursts where upper-tail coupling across neighboring windows matters, while the $t$ component improves robustness to heavy-tailed variability. On SMD, Gumbel & Clayton performs best on both metrics, which is plausible for this heterogeneous dataset: different machines can exhibit different extreme regimes, and combining upper-tail (Gumbel) and lower-tail (Clayton) dependence provides a better match to asymmetric extreme patterns across correlated windows.

Overall, this ablation highlights a key advantage of our pair-copula construction: Copula-SVI is not tied to a single global copula family. Instead, it supports edge-wise and mixed copula strategies that adapt to heterogeneous and asymmetric dependence, while preserving the same edge-decomposable objective for scalable learning of $\log c(\mathbf{U}; \psi)$.

