# OpenReview forum: "Copula-SVI: Vine-Copula Variational Inference with Stein Refining for Instance-Level Correlation Capturing"
_ICML.cc/2026/Conference — ICML 2026 regular_

### Official Review · Reviewer_AeKM · 2026-03-08

**Soundness:** 2
**Presentation:** 3
**Significance:** 3
**Originality:** 2
**Overall Recommendation:** 4
**Confidence:** 3

**Summary:**

This paper introduces a variational method for learning the posterior of latent variables under the dependence setting, where a factorized assumption is inappropriate. A vine copula is introduced to induce dependence in the posterior over the latent variables, where Stein variational gradient descent (SVGD) is then used. The method is then applied in some interesting scenarios including time series and constrained clustering.

**Compliance With Llm Reviewing Policy:**

Affirmed.

**Final Justification:**

While the authors' method can be viewed as an alternative method of training (based on SVGD + the aforementioned approximations) of the specific variational posterior of Tran et al. (2015), their recent rebuttal highlights the significant amount of extra work needed to make it scale to a large number of instances. I am happy to raise my score to 4, conditional on the inclusion of the discussion in their recent rebuttal regarding the connections (and improvements) to Tran et al. (2015), as well as the discussion on the use of a parametric variational family.

**Key Questions For Authors:**

1.	I have some doubts around the discussion in Section 1 on SVGD and projection to factorized approximations, and in Section 3.1 on "SVGD-based Variational EM approaches". The authors claim that SVGD methods "project back to the same factorized family", but this is not a standard application of SVGD. If I am not mistaken, the purpose of SVGD methods is to avoid specifying a variational family at all, so I found this discussion rather confusing. I cannot find this projection step mentioned in Pu et al. 2017 or any other references. Section 3.1 also reads like a review of standard methods, but there are no references that discuss the use of SVGD in this `variational EM’ literature. The only reference is Liu & Wang, 2016 which introduces the original SVGD algorithm and does not discuss this. Perhaps I am missing something, but are there any standard references that discuss this? If so, please clarify and it would be good to include this in the paper.

2.    Furthermore, in addition to my previous comment, it is unclear what the benefit of the projection step is. Could you please explain why you would want to project instead of just directly using the output from SVGD?

3.	The connections to Tran et al. (2015) are severely understated. Like this paper, they use vine copulas to couple together a factorized variational posterior over latent variables. They use stochastic gradient descent on the ELBO, which also seems like a much more natural choice than the SVGD + projection approach. Why is the SVGD approach more appropriate than what Tran et al. (2015) suggests? And how different would your method perform in practice? This is quite an obvious comparator that should be included in the experiments as well.

**Limitations:**

Yes.

**Strengths And Weaknesses:**

Strengths:
The paper is quite well-written, and the presentation of the methodology is neat. This is an interesting use of copulas in the challenging setting of time-dependent latent variable models, and the experiments are thorough.

Weaknesses:
In my opinion, this paper does not situate itself accurately among related work, particularly among the field of SVGD and the use of copulae within variational inference. This makes it challenging to assess the novelty and soundness of the method. I also find the motivation generally quite confusing. Please see my questions below for more details.

---

> ### Author Rebuttal · Authors · 2026-03-31
>
> We thank the reviewer for the careful reading and comments. Below we respond point by point.
>
> **Q1: Authors claim SVGD methods "project back to the same factorized family", but this is not a standard SVGD application. What is the projection step's benefit? Why not directly use SVGD output?**
>
> **A1:** Sorry for the confusion, but **we believe you have misunderstood our work deeply**.
>
> It is true that vanilla SVGD does not need an explicit variational family. For each input $X$, it optimizes particles $\lbrace Z^{(m)}\rbrace_{m=1}^M$ to approximate the target posterior $p(Z\mid X)$. However, if it is used to train generative models $p_\theta(X, Z)$, the optimization process has to repeat from scratch at every iteration, which is extremely expensive.
>
> To speed up inference, follow-up works on SVGD mostly take the **amortized SVGD paradigm**, where an inference network $f_\eta$ is introduced and trained to output the desired particles directly, avoiding the expensive from-scratch optimization in vanilla SVGD. Specifically, for a training data $X$ at iteration $t$, we first use $f_\eta$ to output initial particles $Z^{(m,0)}=f_{\eta_t}(X,\xi_m)$, then use SVGD to refine them by $T$ steps, obtaining $\lbrace Z^{(m,T)}\rbrace_{m=1}^M$, where $\xi_m$ represents Gaussian noise; and $T$ is the number of updating steps, which is often set to 1 or 2. Given the SVGD-refined particles $\lbrace Z^{(m,T)}\rbrace_{m=1}^M$, we then use them to supervise the training of inference network parameter $\eta$ via regression, which is the so-called projection step.
>
> Our work also follows the amortized SVGD paradigm to inherit its efficiency, and that's why we need the projection step.
>
> **Q2: Cannot find the projection step in Pu et al. 2017. Section 3.1 reads like a review of standard methods.**
>
> **A2:** In Pu et al. (2017), Section 2.3 introduces an inference network $z_{jn}=f_\eta(x_n,\xi_{jn})$, and this Eq. (7) accounts for the updating of inference network, which is the projection step called in our paper.
>
> Section 3.1 is a rewrite of amortized SVGD by grouping all latent variables $\{{\mathbf{z}}_i\}_{i=1}^N$ into a matrix ${\mathbf{Z}}$. The matrix-form is used to help reveal the limitations of previous amortized SVGD methods.
>
> **Q3: The connections to Tran et al. (2015) are understated. Why is SVGD more appropriate? And how different in practice?**
>
> **A3:** Thank you for the chance to clarify this point. A copula variational family generally takes the form
> $$q({\mathbf{z}}; {\mathbf{\lambda}}, {\mathbf{\eta}})=\left[\prod_{i=1}^d q(z_i; {\mathbf{\lambda}})\right] \cdot c\left(Q(z_1; {\mathbf{\lambda}}), \cdots, Q(z_d; {\mathbf{\lambda}}); {\mathbf{\eta}} \right).$$
>
> In Tran et al., it takes a re-parameterization paradigm as used in VAE. Since it is unknown how to re-parameterize a sample from copula distribution, they **only re-parameterize samples from the marginal distribution $q(z_i; \lambda)$,  leaving the copula term $c(\cdot; {\mathbf{\eta}})$ unconsidered.** Hence, although a copula family is used, the training process is largely the same as that using a factorized Gaussian posterior. It only considers the correlation in the KL term of the ELBO, while leaving  the more important likelihood-term term still independent. By contrast, our method circumvents this issue by taking the Stein approach.
>
> Moreover, Tran et al. only models the **correlation among dimensions within an instance**, while our method targets the **correlation among  instances**. If the method of Tran et al. is directly applied to capture the instance-level correlation, a small dataset (e.g. 1000 instances) could make its complexity unacceptable. Thanks to the development of novel vine-decomposition and theoretically-grounded methods like edge minibatching, level-wise sampling, our method can easily scale to model instance-level dependence.
>
> To see this more clearly, we did an extra experiment on a mixed-Gaussian synthetic task ($D=100$). Our method achieves much lower posterior error, while Tran et al. is much slower at test time and also suffers from mode collapse.
>
> |Methods|Negative ELBO ↓|MMD to True Posterior ↓|Test Inference Time (per 1000 samples) ↓|
> |-|-|-|-|
> |Mean-field|245.8|4.85|0.08s|
> |Tran et al.|198.4|2.73|~3200s|
> |Ours|162.3|0.35|1.45s|
>
> **Q4: This paper does not situate itself accurately among related work, particularly SVGD and copula within VI.**
>
> **A4:** Due to space limits, we may not position our work well. As noted in A1, our method follows the amortized SVGD paradigm for efficiency. Prior amortized SVGD uses independent variational posteriors, while we are the first using a copula to capture cross-instance correlation. To ensure scalability, we develop theoretically-grounded techniques like level-wise sampling, and unbiased edge-minibatched gradient estimators. Regarding copula VI, prior methods either cannot get correlated re-parameterized samples or lack scalability to capture instance-level correlation. See A3 for details.

---

> > ### Author Rebuttal · Reviewer_AeKM · 2026-04-02
> >
> > Thank you very much for your detailed response - this has helped clarify the purpose and motivation for the paper. I apologize for any misunderstandings in my earlier review, and appreciate the authors' patience.
> >
> > Unfortunately, I still have some issues with the way the paper is framed.
> >
> > **SVGD and Amortization**
> >
> > Thank you for correcting my misunderstanding. It might be worthwhile to include Feng et al. (2017) as a reference, as this paper uses the term "amortized SVGD" explicitly which was quite helpful in clarifying.
> >
> > I think the interpretation as an "amortized SVGD" approach needs to be handled with a bit of care in the setting of this paper, especially because (6) is still quite a restrictive model due to the parametric vine copula, and quite far away from the nonparametric flexibility of SVGD. This is quite different to the setting of Feng et al. (2017) and related work, where it is assumed that the amortized model is flexible enough to approximate the SVGD particles. This was the source of my original confusion.
> >
> > In my own view, because of the above, the authors' method appears to be an alternative way of training an amortized but still parametric variational family given by (6), using SVGD as a tool for training. In my opinion, this is quite different to approximating SVGD with amortization as in previous works. I am still uncertain about the advantages of this training style - please see my comment below regarding comparisons with the training method of Tran et al. (2015).
> >
> > **Comparison to Tran et al. (2015)**
> >
> > Thank you for your detailed comparison. However, I would like to push back on some of the potentially inaccurate statements in the response about the method of Tran et al. (2015). This inaccuracy also makes it a bit difficult to interpret what is carried out in the additional experiments.
> >
> > 1. The authors claim that the copula term is not considered in the reparametrization trick. To my understanding, this is incorrect - **in Section 3.1** of Tran et al. (2015), they explicitly state a reparametrization trick in terms of uniform random variables, which is then derived in **Appendix 1**. This is quite a well-known transformation based on the probability integral transform.
> >
> > 2.  It is claimed that correlation among instances is not modelled. This is more subtle, but again I believe it is incorrect. In **equation (3)** of Tran et al. (2015), the latent variable is presented in full generality, where $z$ is simply a vector of latent variables. This would encompass the setting where $z$ contains both dimension and instance indices, as there is no statement of likelihood factorization assumptions prior to this.
> >
> > 3.  The authors claim that the complexity would be unacceptable if instance correlation is also captured, and (I think) claim that their novel vine-decomposition will help resolve this. Perhaps I misunderstood the authors' comment, but **vine copulas** are specifically mentioned in **Section 2.2.1** of Tran et al. (2015), where they specifically leverage this for more scalable gradients. This is very much the same motivation as provided by the authors.
> >
> > **Summary**
> >
> > I acknowledge the authors' contributions in terms of edge mini-batching and level-wise sampling, but I would argue that the authors' method can be viewed as an alternative method of training (based on SVGD + the aforementioned approximations) of the specific variational posterior of Tran et al. (2015). I have updated my score thanks to the authors' clarifications, but unfortunately cannot give a higher score due to strong overlap with Tran et al. (2015) and a persisting inaccurate contextualisation within related work.
> >
> > Feng, Yihao, Dilin Wang, and Qiang Liu. "Learning to draw samples with amortized stein variational gradient descent." _arXiv preprint arXiv:1707.06626_ (2017).

---

> > > ### Author Response · Authors · 2026-04-04
> > >
> > > We sincerely thank you for the follow-up and apologize for the imprecision in prior response. We agree with you regarding the theoretical formulation in Tran et al. (2015). We will credit their contribution in our final version. However, we must emphasize the fundamental differences between these two works.
> > >
> > > Before diving into algorithmic mechanisms, we first draw your attention to the empirical experiments in two works:
> > >
> > > * Tran et al. validated their approach on two synthetic datasets, with correlation dimensions **2 and 10**, respectively.
> > > * In contrast, our work successfully models instance-level correlation over **566,000 instances**.
> > >
> > >
> > > This **$50,000\times$ gap of correlation dimensions modelled** is not a matter of hardware or implementation. It stems from fundamental algorithmic limitations in Tran et al.'s **reparameterization-based** framework. Below, we explain why this gap exists.
> > >
> > > **1. The Flawed Lower Bound Challenge**
> > >
> > > Exact sampling from a full instance-level vine costs $O(N^2)$, intractable for large $N$. While one might assume Tran et al. could also adopt a sparse vine to save compute, their reparameterization framework cannot do so without yielding a **flawed lower bound**. Their Inverse Rosenblatt sampler relies on a deeply nested cascade of recursive conditional CDF inversions $z_k=F_{k|1:k-1}^{-1}(u_k|z_1,\dots,z_{k-1};\eta)$. If parameters are truncated to only sparse edges $\eta_{sparse}$, the generated samples $\tilde{z}=T_{Rosenblatt}^{-1}(u; \eta_{sparse})$ will deviate from those drawn from a true full vine. The true ELBO requires full model: $\mathcal{L}=\mathbb{E}_ q[\log p(x,z)-\log q(z;\eta)]$. However, relying on the truncated sampler alters the objective to a corrupted surrogate: $\tilde{\mathcal{L}}=\log p(x,\tilde{z})-\log q(\tilde{z};\eta_{sparse})$. Since $\tilde{\mathcal{L}}$ uses heavily distorted samples, it entirely deviates from the true target. When evaluating the gradient $\nabla_{\eta_{sparse}}\tilde{\mathcal{L}}=\nabla_{\tilde{z}}\tilde{\mathcal{L}}(\tilde{z}(u;\eta_{sparse}))\cdot\nabla_{\eta_{sparse}}\tilde{z}(u;\eta_{sparse})$, structural collapse occurs: omitted edge gradients vanish, and retained edge gradients are misdirected by the wrong objective $\tilde{\mathcal{L}}$. Consequently, the model is forced to learn a distorted joint distribution, failing to capture the true dependencies.
> > >
> > > * **Our Solution:** We draw initial samples from a Sparse Vine (Eq. 16), reducing cost to $O(|S_t|)$. We then use SVGD refinement to actively inject missing dependencies back into particles. As proven in Thm 3.4, SVGD acts as a dynamic bias corrector, achieving $O(|S_t|)$ sampling without sacrificing full-vine expressiveness.
> > >
> > > **2. The Recursive Gradient Bottleneck**
> > >
> > > In Tran et al. gradients must backpropagate through the Inverse Rosenblatt sampling path. This involves differentiating through hundreds of layers of nested numerical root-finding and conditional CDFs, which is slow and numerically unstable.
> > >
> > > * **Our Solution:** We decouple sampling from optimization. In the projection update (Eq. 4), we maximize the analytical log-density using refined particles. **Our gradients never flow through the Inverse Rosenblatt sampler**, ensuring efficient and stable optimization of deep vines.
> > >
> > > **3. Dense Evaluation vs. Unbiased Edge-Minibatching**
> > >
> > > Even if efficient reparameterization were achieved, Tran et al.'s framework still faces training difficulties. Evaluating full vine density requires $O(N^2)$ pair-copula evaluations per iteration. While they mention truncated vines to reduce complexity, they lack mechanism for unbiased stochastic subsampling during optimization.
> > >
> > > * **Our Solution:** We introduce Horvitz-Thompson unbiased edge-minibatching (Eq. 12) with Variance-Optimal Level-Wise Sampling (Thm 3.3). This allows us to update parameters of a **full-vine copula** using only a small bounded edge budget $O(|S_t|)$ per iteration while maintaining an unbiased gradient estimator.
> > >
> > > ---
> > > **Q: The method appears to train an amortized but still parametric variational family... uncertain about the advantages of this style.**
> > >
> > > **A:** We fully agree with your perspective that our method uses a parametric variational family, whereas amortized SVGD (e.g., Feng et al., 2017) employs a non-parametric (implicit) model, appearing more flexible.
> > >
> > > However, **such implicit models only show superiority when used to capture correlations among dimensions within an instance.** To model correlations across $N$ instances,  all $N$ instances must be fed into the inference network as joint input. Since $N$ is generally very large (thousands to millions), this would lead to an intractable input space.
> > >
> > > By restricting the distribution to the parametric vine-copula family (Eq. 8), we decompose this distribution into bivariate factors. Thus, our neural network only needs to process at most two instance endpoints simultaneously, providing a viable path for instance-level correlation modeling.

---

### Official Review · Reviewer_WwWM · 2026-03-12

**Soundness:** 4
**Presentation:** 4
**Significance:** 4
**Originality:** 4
**Overall Recommendation:** 6
**Confidence:** 5

**Summary:**

This paper introduces Copula-SVI, a variational inference framework specially designed for scenarios where there is correlation in the data. The article claims to consider the concept of instance-level dependency modelling within variational inference, which is often ignored by standard approaches that assume independence across data points. They propose to model marginal and dependency using a copula trick and introduce a family of copula-augmented variational posteriors. This method uses a vine-copula decomposition, which represents high-dimensional dependencies as a set of pairwise copulas organized in multiple levels. Further, to reduce computation costs, they consider a minibatch of vine edges to obtain an unbiased gradient estimator and propose a method to reduce its variance by allocating the edge budget across vine levels. These innovations are theoretically supported as well. The approach is further improved using Stein Variational Gradient Descent (SVGD) with Rosenblatt transformation to refine samples so that the approximate posterior better matches the true posterior while preserving learned dependencies. Finally they show that the algorithm has a vanishing initialization bias and present a quantification of the approximation gap via omitted-edge energies.

 Experimental results on constrained clustering, time-series anomaly detection, and time-series forecasting show that Copula-SVI consistently outperforms existing structured variational inference methods and scales better when capturing higher-order relationships among instances. These findings highlight the importance of explicitly modelling instance-level correlations in probabilistic inference for complex datasets.

**Compliance With Llm Reviewing Policy:**

Affirmed.

**Key Questions For Authors:**

In my opinion, this is a very nice work and I do not think of any additional questions presently to answer.

**Limitations:**

yes

**Strengths And Weaknesses:**

I found this is a very nice paper,
- bringing together vine copula+clever sampling for gradient estimate+SVGD refinement altogether to provide a new flexible VI family;
- new estimates, sampling schemes have been justified via theoretical proofs;
- simulation study section is illustrative;
- I am satisfied with their discussions of limitations.

---

> ### Author Rebuttal · Authors · 2026-03-31
>
> We sincerely thank the reviewer for the very positive and encouraging assessment of our paper. We are glad that the reviewer found the proposed combination of vine copulas, scalable gradient estimation, and SVGD refinement to be novel, technically sound, and well justified. We also greatly appreciate the recognition of our theoretical analysis, experimental study, and discussion of limitations. Thank you again for the strong support and for the careful reading of the paper. In the revision, we will further improve the exposition and incorporate the constructive feedback from the other reviewers.

---

> > ### Author Rebuttal · Reviewer_WwWM · 2026-04-03
> >
> > Thanks for your effort and work to clarify other reviewers concerns. I still keep my same positive assessment of the paper.

---

### Official Review · Reviewer_Hidd · 2026-03-17

**Soundness:** 3
**Presentation:** 3
**Significance:** 2
**Originality:** 2
**Overall Recommendation:** 5
**Confidence:** 4

**Summary:**

The paper proposes a new way to model the correlation between the latent variable associated with each data point. In particular, the paper uses a mean-field variational family augmented with a copula and then refines the quality of the samples by applying multiple steps of Stein variational gradient descent (SVGD). Since the copula models the conditioning over the whole data set, learning the copula exactly is intractable. Therefore, the paper proposed a biased scheme that only uses minibatch subsets. The bias is supposedly corrected thanks to the SVGD refinements. As a result, the proposed scheme enables learning correlations between data points and can therefore be used for clustering. Evaluations are based on clustering accuracy and time-series forecasting error.

**Compliance With Llm Reviewing Policy:**

Affirmed.

**Final Justification:**

The technical concerns have been addressed through additional experimental results. There are still some outstanding questions about the motivations for some of the design choices, which honestly still feel a bit *ad hoc*. But I believe the paper does meet the bar for publication at ICML.

**Key Questions For Authors:**

in progress

**Limitations:**

in progress

**Strengths And Weaknesses:**

## Strengths
* Modeling the cross-datapoint correlation in the amortized variational inference setting is an important and challenging problem.
* The proposed method relaxes the parametric assumptions made in Hot-VI [1] by leveraging vine copulas. Coming up with an efficient and scalable algorithm for this modification is non-trivial.

## Weaknesses
* The methodology feels a bit *ad hoc*. First, the $O(N^2)$ cost of sampling from a vine copula is circumvented by using a biased strategy. The submission mitigates the bias by applying SVGD refinements to the samples. But applying SVGD is not a fundamental fix in the sense that any initial variational approximation $q$ can be refined by applying SVGD. I would like to see how serious this bias is and how much work SVGD is doing to neutralize this. Also, why SVGD? How about using Hamiltonian Monte Carlo similarly to [3]? Some justification for this design choice would be needed.
* This leads to a more critical problem with the evaluations. Since the other VAE baselines can also be refined via SVGD, but do not enjoy this treatment, the evaluation is a bit unfair. Furthermore, it is currently unclear whether the performance improvement is due to SVGD or to the use of vine copulas.
* Theorems 3.4 and 3.5 are not very insightful and feel a bit too decorative. For instance, the statement of Theorem 3.4 is intuitive. The analysis itself is also not interesting, as it relies on strong assumptions that pretty much imply the conclusion. I would recommend simply removing it. Similarly, Theorem 3.5 also relies on the strong assumption that a well-defined Fisher information exists. Even then, I am not sure how to interpret the approximation gap established in Theorem 3.5. Is it really negligible? When is it not negligible? Can the results be supported with empirical evidence? To be honest, I feel the paper meets the bar without these Theorems. So I wouldn't insist on presenting/improving these results.
* The discussion about related works could be improved.
    * For instance, the way SVGD is used to refine the samples is reminiscent of the line of work that uses MCMC to refine variational approximations. For example, [2,3].
    * The variance reduction strategy in Section 3.4.2 is a well-studied strategy in stochastic optimization called importance sampling. It has also been considered in VI in [4,5,6,7]. (There might be more papers that do this in VI, but I didn't search for this too hard.) Drawing connections with this literature would strengthen the paper.
    * In addition, the discussion on past works in copula variational inference should be expanded. Right now, the related works section only mentions [8] as a method that hierarchically constructs a variational family. There should be a clarification that [8] in particular attempted to generally model all correlations, whereas this paper focuses on modeling the correlations among local variables only. Furthermore, copula VI methods have some history, and I think other follow-up works, such as [9,10,11,12], should be added and discussed. Some discussion on the gradient estimators would also be interesting.

## Additional Comments
* Line 40 "In the standard i.i.d. setting": The term "i.i.d. setting" is misleading/ambiguous, and I recommend against using it. It would be more appropriate to say "*conditionally* independent" or "exchangeable" setting.
* Line 46 "is a natural choice": I don't agree that this is natural. The fact that the data is c.i.d. does not mean the latent variable posterior is well approximated by a mean-field family. How about "is a common choice due to computational efficiency"?
* Line 99 "standard i.i.d. regime": Again, please do not say i.i.d. here.
* Section 3.1: I think this would be better moved to the background section so that readers familiar with SVGD can skip this part.
* Eq (4): Where was $\Phi$ defined? Is it a typo for $\phi$?
* Line 128 "data are i.i.d.": Are you assuming *identically* distributed data here?
* Eq (7): Is $\Phi$ supposed to be $\phi$ in $q_{\Phi}$?
* Theorem 3.1 to Line 218: Wouldn't this belong to the background section?
* Eq (256) "introduces a bias that persists": Can the bias be explained more mathematically using equations?


1. Xiao, J., Su, Q., & Yuan, Z. HoT-VI: Reparameterizable variational inference for capturing instance-level high-order correlations. *NeurIPS*.
2. Salimans, T., Kingma, D., & Welling, M. (2015, June). Markov chain monte carlo and variational inference: Bridging the gap. *ICML*.
3. Hoffman, M. D. (2017, July). Learning deep latent Gaussian models with Markov chain Monte Carlo. *ICML*.
4. Needell, D., & Ward, R. (2016, May). Batched stochastic gradient descent with weighted sampling. In *International Conference Approximation Theory*.
4. Csiba, D., & Richtárik, P. (2018). Importance sampling for minibatches. *JMLR*.
5. Zhao, P., & Zhang, T. (2015, June). Stochastic optimization with importance sampling for regularized loss minimization. *ICML*.
6. Hanzely, F., & Richtárik, P. (2019, April). Accelerated coordinate descent with arbitrary sampling and best rates for minibatches. *AISTATS*.
7. Gower, R. M., Richtárik, P., & Bach, F. (2021). Stochastic quasi-gradient methods: Variance reduction via Jacobian sketching. *Mathematical Programming*.
8. Tran, D., Blei, D., & Airoldi, E. M. (2015). Copula variational inference. *NeurIPS*.
9. Han, S., Liao, X., Dunson, D., & Carin, L. (2016, May). Variational Gaussian copula inference. *AISTATS*.
10. Smith, M. S., Loaiza-Maya, R., & Nott, D. J. (2020). High-dimensional copula variational approximation through transformation. *JCGS*.
11. Fu, Y., Smith, M. S., & Panagiotelis, A. (2025). Vector Copula Variational Inference and Dependent Block Posterior Approximations. *JCGS*.
12. Smith, M. S., & Loaiza-Maya, R. (2023). Implicit copula variational inference. *JCGS*.

---

> ### Author Rebuttal · Authors · 2026-03-31
>
> We thank the reviewer for the careful reading and comments. Below we respond point by point.
>
> **Q1: The submission mitigates the bias of sampling from a vine copula by applying SVGD refinements to the samples. I would like to see how serious this bias is and how much work SVGD is doing to neutralize this.**
>
> **A1:** Sorry for the lack of clarity in our presentation. To reduce the sampling cost, we do sample from a sparse vine, but the obtained sample is only used as **initialization particles** for the SVGD refining process. Given these initialized particles, we then refine them by applying SVGD to move them toward better representing the true posterior $p(Z|X)$. Then, we use the refined particles to supervise the updating of parameters **in the full-vine posterior** via regression, rather than the sparse-vine posterior. Thus, although the initial particles are probably biased, they have later been corrected by SVGD to reflect the exact posterior distribution, and the corrected particles are further written back to the parameters of full-vine copula. Therefore, despite the bias in initial particles, the full-vine copula is always aiming to approximate the true posterior distribution.
>
> Moreover, to make the initial particles as accurate as possible, we develop an effective method to keep the most informative edges, thereby reducing the effect of not considering all edges to minimum. Table 1 directly shows the practical effect of sampling from a sparse vine in our method. When the number of edges in the sparse-vine is reduced, the final performance degrades only gradually, suggesting that the approximation introduced at initialization has only a limited effect.
>
> Table 1: Varying sparse-vine budget on MNIST (10-order, fixed edge minibatch)
> |Edges|ACC|NMI|ARI|
> |-|-|-|-|
> |32|98.0±0.3|95.5±0.4|95.9±0.3|
> |64|98.4±0.2|95.9±0.2|96.3±0.2|
> |128 (default)|98.6±0.2|96.2±0.3|96.7±0.1|
> |256|98.7±0.2|96.3±0.2|96.8±0.1|
>
> **Q2: Why SVGD instead of Hamiltonian Monte Carlo (HMC)?**
>
> **A2:** HMC introduces strong randomness. With few steps, it is unreliable (particles may worsen or stagnate); requiring many steps for mixing makes it too expensive for per-iteration refinement. Conversely, SVGD constructs deterministic updates from the KL divergence. It directly and stably pushes particles toward the target, achieving strong refinement in very few steps, perfectly fitting our efficiency needs.
>
> **Q3: Other VAE baselines can also be refined via SVGD. Is the evaluation unfair? Is the improvement due to SVGD or vine copulas?**
>
> **A3:** Thank you for pointing this out. Other VAE baselines can also be refined by SVGD, but they are not essentially compatible with the same SVGD treatment as our method. Factorized VAEs assume instance independence; thus, cross-instance dependence gained via SVGD is lost when fitting particles back to the posterior. Reparameterization methods (e.g., HoT-VI) lack a decomposable structure. To retain SVGD's refined dependence, they must jointly refit all dependence parameters, which is intractable compared to our efficient edge-wise updates.
>
> The table below shows why the gain is not from SVGD alone. Adding SVGD to a factorized posterior helps only slightly, because the factorized family still cannot keep the cross-instance dependence improved by SVGD. Our method performs much better because the vine-copula posterior can preserve that dependence.
> |Method|ACC|NMI|ARI|
> |-|-|-|-|
> |Factorized|96.5±0.2|91.4±0.3|92.5±0.5|
> |Factorized+SVGD|96.8±0.2|91.8±0.3|93.1±0.4|
> |**Copula+SVGD (Ours)**|**98.6±0.2**|**96.2±0.3**|**96.7±0.1**|
>
> **Q4: Theorems 3.4 & 3.5 rely on strong assumptions. Is there empirical evidence?**
>
> **A4:** Thanks for raising this. Theorem 3.4 addresses the limiting case (strong refinement erases sparse initialization effects), while Theorem 3.5 bounds finite-step approximation error via omitted-edge energies (weaker omitted edges = smaller gap). Empirical results show the same trend: increasing the sparse-vine budget (32 to 256) drops omitted-edge energy from 1.84 to 0.23, steadily improving performance (Table 1), matching Theorem 3.5. We acknowledge these are not the core contributions and will condense their presentation.
>
> **Q5: The discussion about related works could be improved.**
>
> **A5:** Relative to MCMC literature, we use a posterior-improvement step but learn an explicit posterior preserving cross-instance dependence. Section 3.4.2 relates to importance-sampling via vine-specific weighted allocation to keep the objective trainable. Crucially, prior copula VI relies on VAE-style re-parameterization which leaves copula terms inactive, and mainly models dependence among dimensions within an instance. We instead use Stein updates to actively capture inter-instance correlation. Since direct copula does not scale well in this setting, we introduce vine decomposition and scalable vine learning to make training feasible. We will incorporate these discussions more explicitly in the final version.

---

> > ### Author Rebuttal · Reviewer_Hidd · 2026-04-04
> >
> > I thank the authors for their response. I appreciate the additional experiments, and in light of this, I will raise my score.
> >
> > > Q1: The submission mitigates the bias of sampling from a vine copula by applying SVGD refinements to the samples. I would like to see how serious this bias is and how much work SVGD is doing to neutralize this.
> >
> > For this, I was specifically asking for an ablation of the effect of SVGD.
> >
> > > > Q2: Why SVGD instead of Hamiltonian Monte Carlo (HMC)?
> > >
> > > A2: HMC introduces strong randomness. With few steps, it is unreliable (particles may worsen or stagnate); requiring many steps for mixing makes it too expensive for per-iteration refinement. Conversely, SVGD constructs deterministic updates from the KL divergence. It directly and stably pushes particles toward the target, achieving strong refinement in very few steps, perfectly fitting our efficiency needs.
> >
> > This answer doesn't quite make sense to me. HMC may not always be *geometrically* ergodic, but it is going to be ergodic no matter what under very weak conditions. Therefore, applying HMC will always improve the approximation accuracy of any set of particles (even if it's a single particle).

---

> > > ### Author Response · Authors · 2026-04-06
> > >
> > > We sincerely thank the reviewer for the recognition of our additional experiments and the willingness to raise the score.
> > >
> > > **Q1: For this, I was specifically asking for an ablation of the effect of SVGD.**
> > >
> > > **A1:** We sincerely thank the reviewer for the valuable suggestion. To explicitly demonstrate the effect of the SVGD module, we conduct an ablation study on the SVGD refinement steps ($T$) for both the constrained clustering (Table 1) and time series anomaly detection (Table 2) tasks.
> > >
> > > Table 1: Impact of SVGD Refinement Steps ($T$) on Constrained Clustering Performance for MNIST and Fashion-MNIST (10-order Copula-SVI)
> > >
> > > | Dataset        |              | MNIST       |              |              | fMNIST       |              |
> > > | :- | :- | :- | :- | - | - | - |
> > > | Steps           | ACC          | NMI          | ARI          | ACC          | NMI          | ARI          |
> > > | $T=0$           | 97.6±0.3     | 93.8±0.5     | 94.2±0.6     | 81.6±0.8     | 74.2±0.6     | 68.6±0.6     |
> > > | $T=1$           | 98.1±0.4     | 95.1±0.4     | 95.5±0.3     | 82.7±0.5     | 74.9±0.5     | 68.9±0.5     |
> > > | $T=2$           | 98.3±0.2     | 95.6±0.2     | 96.0±0.2     | 83.1±0.6     | 75.2±0.4     | 69.0±0.5     |
> > > | $T=3$           | 98.4±0.3     | 95.8±0.4     | 96.4±0.4     | 83.6±0.4     | 75.1±0.4     | 69.3±0.3     |
> > > | $T=4$           | 98.5±0.2     | 96.0±0.3     | 96.5±0.2     | 83.7±0.3     | 75.4±0.3     | 69.4±0.2     |
> > > | $T=5$ (default) | **98.6±0.2** | **96.2±0.3** | **96.7±0.1** | **83.8±0.3** | **75.6±0.2** | **69.5±0.2** |
> > >
> > > Table 2: Impact of SVGD Refinement Steps ($T$) on Time Series Anomaly Detection Performance (10-order Copula-SVI)
> > >
> > > | Dataset | SMAP |     |  MSL  |   |  SMD  |      |
> > > | :- | :- | :- | - | - | - | - |
> > > | Steps           | F1-score   | ELBO         | F1-score   | ELBO          | F1-score   | ELBO         |
> > > | $T=0$           | 0.8481     | -97.2314     | 0.8912     | -155.3412     | 0.8925     | -70.8123     |
> > > | $T=1$           | 0.8623     | -93.8451     | 0.9084     | -142.1054     | 0.9156     | -67.4521     |
> > > | $T=2$           | 0.8685     | -92.5102     | 0.9167     | -138.4501     | 0.9263     | -65.8190     |
> > > | $T=3$           | 0.8712     | -91.7345     | 0.9198     | -135.2133     | 0.9310     | -64.9214     |
> > > | $T=4$           | 0.8741     | -91.2189     | 0.9231     | -133.5610     | 0.9352     | -64.2105     |
> > > | $T=5$ (default) | **0.8753** | **-90.9828** | **0.9245** | **-132.6568** | **0.9371** | **-63.7555** |
> > >
> > >
> > >
> > > As shown in the experimental results above, we observe two main phenomena:
> > >
> > > - Introducing just one step of SVGD refinement yields a substantial performance boost. This demonstrates that SVGD quickly provides a clear gradient direction to improve particle quality in very few steps.
> > > - As we continue to increase the SVGD steps, the model performance improves steadily, but the gains gradually shrink and begin to plateau.
> > >
> > > To strike a balance between maximizing performance and maintaining a manageable computational cost, we selected $T=5$ as the default setting in our main experiments. We will incorporate these ablation results and more comprehensive discussions into the final version of our paper.
> > >
> > >
> > >
> > > ---
> > >
> > > **Q2: HMC may not always be *geometrically* ergodic, but it is going to be ergodic no matter what under very weak conditions. Therefore, applying HMC will always improve the approximation accuracy of any set of particles (even if it's a single particle).**
> > >
> > > **A2:** We thank the reviewer for the helpful clarification. We agree that applying HMC can also improve the approximation accuracy of the particles. Our choice of SVGD over HMC is primarily motivated by our core objective: the refined particles are immediately used to explicitly fit a parametric vine-copula posterior. Since we operate under a small refinement budget, what matters is not only whether each particle moves toward a higher-density region, but **whether the particle set quickly exhibits the joint dependence structure** that the copula needs to fit.
> > >
> > > From this perspective, SVGD is better aligned with our objective than HMC. HMC updates particles through **independent Markov transitions**, which effectively improve individual trajectories but do not explicitly coordinate the particles to form the overall joint distribution shape. SVGD instead updates particles jointly: besides moving them toward high-density regions, its **interaction term (kernel repulsion) spreads particles collectively**, so the particle cloud more quickly reflects the global dependence structure needed for copula projection. Furthermore, SVGD acts as a **deterministic gradient flow along the steepest descent direction** of the KL divergence within an RKHS. This strictly avoids the stochastic momentum and potential Metropolis-Hastings rejections inherent to HMC, yielding a much more stable and efficient structural fit within a tightly limited step budget. We will make the motivation for this design choice much clearer in our final version.

---

### Decision · Program_Chairs · 2026-04-30

**Decision:**

Accept (regular)

**Comment:**

The paper develops a scalable structured VI method that models instance-level correlations using a vine-copula posterior, plus SVGD refinement and scalable edge-minibatched training.

While the reviewers praise the richer posterior family, the scalable design, the strong empirical results, and the generally clear presentation, they also raise several concerns, including that the method can feel somewhat ad hoc, especially around sparse-vine sampling, the role of SVGD, fairness versus baselines, and the positioning relative to prior copula/SVGD literature.

The authors' rebuttals added ablations on sparse-vine budget and factorized+SVGD baselines, justified using SVGD instead of HMC, and expanded the comparison to prior copula/amortized-SVGD work, especially Tran et al. (2015).
The reviewers indicated that these responses have addressed most of their criticisms.

Therefore, we have decided to accept the paper for presentation at ICML. We would still recommend that the authors take the reviewers' feedback into account when preparing the camera-ready version.